# A low-cost technique to measure bank erosion processes along middle-size river reaches

Gonzalo Duró[1], Alessandra Crosato[1,2], Maarten G. Kleinhans[3], Wim S. J. Uijttewaal[1]

[1] Department of Hydraulic Engineering, Delft University of Technology, PO Box 5048, 2600 GA Delft, the Netherlands
[2] Department of Water Engineering, IHE-Delft, PO Box 3015, 2601 DA Delft, the Netherlands
[3] Department of Physical Geography, Utrecht University, PO Box 80115, 3508 TC Utrecht, the Netherlands

*Correspondence to*: Gonzalo Duró (G.Duro@tudelft.nl)

We investigate the capabilities of Structure from Motion (SfM) photogrammetry applied with imagery from an Unmanned Aerial Vehicle (UAV) to measure bank erosion processes in middle-size rivers. This technique offers a unique set of characteristics compared to previously used methods to monitor banks, such as high resolution, low-cost and relatively fast deployment in the field. We analyse a 1.2 km restored bank of the Meuse River with complex vertical scarps laying on a straight reach, features that present specific challenges to the UAV-SfM application. We surveyed eight times within a year, combining different photograph perspectives and overlaps to identify an effective UAV flight. The accuracy of the Digital Surface Models (DSMs) was evaluated with RTK GPS points and an Airborne Laser Scanning (ALS) of the whole reach. An oblique perspective with eight photo overlaps and 20 m of cross-sectional ground-control point (GCP) distribution were sufficient to achieve the relative precision to observation distance of ~1:1400 and 3 cm RMSE, complying with the required accuracy. A complementary nadiral view increased coverage behind bank toe vegetation. The GCP footprint across the floodplain proved critical to avoid rotation of straight elongated domains, so improvements to the adopted approach are recommended. Sequential DSMs captured signatures of the erosion cycle such as mass failures, slump-block deposition, and bank undermining. Although this technique requires low water levels and banks without dense vegetation as many others, it is an inexpensive and fast-in-the-field alternative to survey reach-scale riverbanks in sufficient resolution to quantify bank retreat and identify morphological features of the bank failure and erosion processes.

Keywords:    Riverbank erosion monitoring, erosion cycle, restoration, Unmanned Aerial Vehicles (UAV), Structure from Motion (SfM).

## 1 Introduction

Bank erosion is a fundamental process in morphologically active river systems, and much research has been devoted to understanding, quantifying and modelling it from disciplines such as engineering, geomorphology, geology and ecology. River bank erosion involves interconnected physical, chemical and biological processes (e.g., Hooke, 1979; ASCE, 1998; Rinaldi and Darby, 2008), resulting in a complex phenomenon that is difficult to thoroughly understand and predict (e.g.,

Siviglia and Crosato, 2016). Predicting and monitoring bank erosion is necessary for sound river management strategies and also important for both socio-economic problems, such as preventing material losses (e.g., Nardi et al., 2013), and environmental challenges, for instance, promoting habitat diversity through river restoration (e.g., Florsheim et al., 2008) and improving water quality (e.g., Reneau et al., 2004).

Bank erosion can be monitored with different spatial resolutions, time frequencies and accuracies. The techniques that identify the temporal change in vertical bank profiles detect and quantify the different phases of the erosion cycle (Thorne and Tovey, 1981). This characteristic helps distinguishing the factors influencing bank erosion and their relative role in the whole process (e.g., Henshaw et al., 2013). On the other hand, a simple record of sequential mass failure events (see Fukuoka, 1994, for a graph of failure-driven retreat) is sufficient to track rates of local bankline retreat and estimate eroded

volumes, but does not provide further information on the role of single factors governing the bank erosion process. In navigable rivers, for instance, it is important to differentiate the effects of vessel-induced waves from the effects of river flow, as well as those of high flows and water level fluctuations. This requires high spatial resolution and relatively frequent measurements that usually involve expensive equipment and field logistics when monitoring large extensions. In this context, Structure from Motion (SfM) photogrammetry appears a promising low-cost technique to measure bank erosion

processes along extensive distances (Fonstad et al., 2013).

      We investigate whether the resolution, precision and frequency of acquisition of SfM applied with imagery from a low-cost multi-rotor Unmanned Aerial Vehicle (UAV) is capable of monitoring banks at the process scale along a middle-size river reach. In order to do that, we compare the SfM-based Digital Surface Model (DSM) with Real-Time Kinematic (RTK) GPS measurements and Airborne Laser Scanning (ALS), and analyse erosion features in bank profiles considering

the erosion cycle as a reference to distinguish approaches that measure bank erosion. The case study is a 1.2 km straight river reach with complex vertical scarps. This type of linear domain with vertical surfaces represent challenges to the UAV-SfM application, since special UAV paths and camera angles may be needed to capture the bank area and rather aligned GCPs along banks may result in rotated solutions during the model linear transformation.

      The study site is located in the Meuse River near the city of Gennep, the Netherlands, which has recently undergone

a large bank-restoration project. The Meuse is a heavily regulated river used as navigation route to connect the eastern part of Belgium and the Netherlands to the industrial area in the West and the port of Rotterdam. The restoration aims to re-naturalize the previously protected banks, which are now allowed to erode. Here we take advantage of knowledge of the original bank and of a rare event of extremely low water level.

## 2 Framework of analysis

### 2.1 Bank erosion cycle

Bank erosion may consist of three phases (Thorne and Tovey, 1981): fluvial entrainment of near-bank river-bed and bank material, mass failure, and disintegration and removal of slump blocks. These three phases are particularly important for

cohesive banks since their retreat is typically delayed by the protection offered by slump blocks at their toe (Thorne, 1982; Lawler, 1992; Parker et al., 2011). The waste material settles at the bank toe where it remains for a time depending on its resistance to fluvial erosion and on the flow capacity to transport the blocks. In contrast, loose waste material from non-cohesive banks is generally transported away relatively quicker by the river flow, leaving the bank sooner unprotected.

Entrainment of near-bank bed material and the intact bank face occurs once the bank toe is exposed again (e.g., see Clark and Wynn, 2007), which continues until the collapse of the upper bank. Mass failure occurs due to geotechnical instability, which can be triggered by different factors, such as fluvial bank-toe erosion (e.g., Darby et al., 2007) or a rapid drawdown of the river stage (e.g., Thorne and Tovey, 1981; Rinaldi et al., 2004).

Not only the river flow triggers the bank erosion cycle but other drivers can contribute to it as well. For instance,
subaerial processes may weaken the bank and accelerate later fluvial erosion (Lawler, 1992; Kimiaghalam et al., 2015) or also act as direct agent of erosion (Couper and Maddock, 2001). These effects are included in the entrainment phase for the former example and in the mass failure phase for the latter, which respectively promote entrainment or deliver material to the bank toe. Figure 1 illustrates the three phases of erosion in schematic cohesive banks. This representation shows a homogeneous soil which undergoes a continuous cycle of erosion with varying water levels.

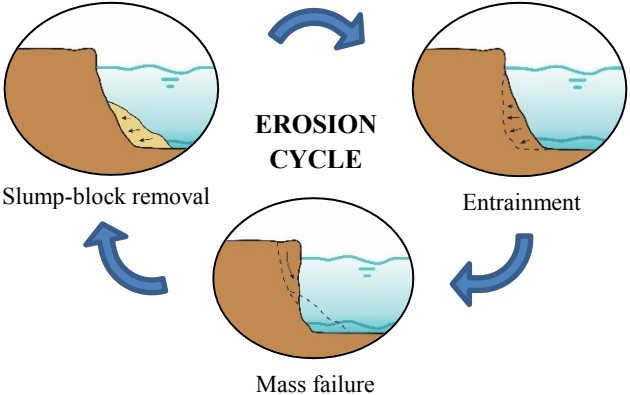

**Figure 1:** Schematic bank erosion phases: Slump-block removal (left), entrainment of bare bank (right) and incipient mass failure (centre)

Bank erosion can be analysed and measured at two different scales, i.e., the fluvial process and the river cross section. The measurement at the process scale considers the bank face disintegration over time with evidence of erosion phases (Fig. 1): the mechanisms of erosion develop and are captured at the vertical dimension of the bank. The measurement of bank erosion at the cross-sectional scale, which can be referred to as bankline retreat, consists of tracking banklines over time. In this case, the focus is on the planimetric changes of the bank edge and estimations of eroded volumes and sediment
yield. The former approach deals with processes and mechanisms (e.g., Rinaldi and Darby, 2008), whereas the latter with landscape development at larger spatial and temporal scales. Bank-erosion studies determine the survey method based on their aims and scales of interest, whereas in turn a given methodology constraints the scope of the findings (Massey, 2001;

Couper, 2004). Thus, it is important to identify capabilities and limitations of each survey technique in the context of river banks, which are inherently steep features with small-scale irregularities independent of the scale of the river.

## 2.2 Techniques to measure bank erosion

Measuring techniques have four essential characteristics: the extent, resolution, precision and frequency of measurements. Extent refers to the area or distance along the river covered by each survey; resolution indicates the distance between surveyed points; precision is the accuracy of position of each surveyed point; and frequency derives from the time interval between consecutive surveys of the same spatial extent or point. The scale of interest may vary among disciplines (e.g., geomorphology, engineering, ecology), so that a diversity of techniques is available with varying spatio-temporal windows of inquiry (Lawler, 1993). Even though the methods currently adopted to measure river bank erosion range from photo-electric erosion pins to terrestrial laser scanning, they have high resolution in either time or space (Couper, 2004; Rinaldi and Darby, 2008).

The methods to determine bankline retreat and to estimate eroded volumes are typical of remote sensing, for instance, ALS and aerial photography. The former technique has typical resolutions of 1 and 0.5 metres, and covers up to hundreds of square kilometres per day. Bailly et al., (2012) indicate decimetre vertical precision, which depends on several factors including beam footprint size, aircraft inertial measuring system, on-board GPS, vegetation cover and filtering technique. ALS has been successfully applied to identify river morphological features, such as bar tops (Charlton et al., 2003) and riffle–pool and step–pool sequences (Cavalli et al., 2008). In addition, sequential ALSs were used to quantify volumes of eroded banks to subsequently estimate pollutant loads, achieving reasonable results for those aims (Thoma et al., 2005). However, banks are particularly steep areas where this technique tends to increase the elevation uncertainty (Bangen et al., 2014). Therefore banks are regions where lower ALS accuracies are expected compared to horizontal and flat areas.

Aerial photography has also been applied to measure bank migration, which is a useful source of information, especially if historical imagery is available over extended periods of time. Yet, it provides only limited information on bank heights. Thus, this planform survey technique requires other methods to estimate eroded volumes. For example, photogrammetry can serve to quantify volumetric changes from overlapping photographs (Lane et al., 2010); or ALS may provide recent topographic elevations to reconstruct past morphologies (Rhoades et al., 2009). Bank retreat can also be estimated through other approaches, such as those described by Lawler (1993), that include planimetric resurveys for intermediate timescales (years) and sedimentological and botanical evidence for long timescales (centuries to millennia).

Measuring bank erosion at the process scale involves measuring the evolution of the vertical bank profile over time and several techniques are currently available to that end. Traditional methods include erosion pins and repeated cross-profiling, which provide two-dimensional information with resolutions that respectively depend on the number of pins and points across the profile (Lawler, 1993). Erosion pins are simple and effective, but their accuracy may be affected by several factors, such as subaerial processes (Couper et al., 2002). More advanced versions are the photo-electric erosion pins that automatically track the bank face during different erosion phases (Lawler, 2005). Cross-profiling can be done with GPS or

total stations with point accuracies of a few centimetres or millimetres, yet with spatial and temporal resolutions that may not be sensitive to very localized or intermittent erosion (e.g., Brasington et al., 2000).

Bank geometries can currently be surveyed with their three-dimensional complexity through a number of techniques whose geomorphic applications are broader than bank erosion studies: terrestrial photogrammetry, Terrestrial Laser Scanning (TLS), boat-based laser scanning and SfM photogrammetry. Terrestrial photogrammetry has shown detailed bank representations, with approximate resolutions of 2 cm and precision within 3 cm, covering up to 60 metres of banks (Barker et al., 1997; Pyle et al., 1997). Yet, this method can be labour-intensive and requires an accessible bank (Bird et al., 2010), known camera positions and sensor characteristics, ground-control points, among other considerations (Lane, 2000). TLS has shown detailed erosion patterns from sequential surveys, with millimetre resolutions, which in practice are usually reduced to 2–5 centimetres, and approximate final accuracies of 2 cm (Resop and Hession, 2010; Leyland et al., 2015). O'Neal and Pizzuto (2011) proved the advantages of 3D TLS in capturing patterns (e.g., overhanging blocks) and quantifying eroded volumes over 2D cross-profiling. Even though TLS could cover thousands of meters, in practice the extents are generally smaller due to accuracy decrease, large incidence angles, occlusion, etc. (Telling et al., 2017), so several scans are necessary to measure long distances. For instance, Brasington et al. (2012) surveyed a 1 km river reach scanning every 200 m along the channel. An alternative boat-based laser scanning can continually survey banks with comparable resolutions and accuracies to those of TLS, with great time reduction but involving other field logistics, resources and post-processing (Alho et al., 2009).

SfM photogrammetry has been applied to measure banks to show its potential use as survey technique with different sensors and processing systems (Micheletti et al., 2015; Prosdocimi et al., 2015). Micheletti et al. (2015) indicated root mean square errors (RMSE) within 7 cm, when combining a 5MP smartphone or a 16MP reflex camera with either PhotoModeler or 123D Catch processing systems. Prosdocimi et al. (2015) identified eroded areas of a collapsed riverbank and computed eroded and deposited volumes with a precision comparable to that of TLS. Bangen et al., (2014) matched the resolution and practical extent of this technique to those of TLS, when SfM photogrammetry is used to survey river topography through aerial platforms (e.g., Fonstad et al., 2013). The relatively recent and fast development of UAV technology to take airborne photographs has greatly expanded the applications of SfM photogrammetry (Eltner et al., 2016). Recently, SfM has been applied to quantify bank retreat at streams and small rivers with a fixed-wing UAV along several kilometres with 12 cm resolution (Hamshaw et al., 2017). This study showed the UAV-SfM capabilities to produce extensive 2.5D DSM from a 100 m high nadiral view, which achieved 0.11 m mean error and 0.33 m RMSE compared to TLS. However, this work generated DSMs similar to those of ALS, which allow for volume computations and bankline retreat, but did not use the full 3D capacities to investigate undermined banks or identify erosion processes.

Applications of this combined technology span in scale and complexity, covering glacial dynamics (Immerzeel et al., 2014), landslides (Turner et al., 2015), agricultural watersheds (Ouédraogo et al., 2014), fluvial topography (Woodget et al., 2015), etc. The accuracy achieved relative to the camera-object distance for the mentioned diverse settings was approximately 1:1000, with distances ranging from 26 to 300 m and different cameras, lighting conditions and surface types.

Interestingly, this precision was also found for terrestrial SfM photogrammetry at different scales by James and Robson (2012). However, other experiences showed lower accuracies, e.g., ~1:200 for moraine-mound topography (Tonkin et al., 2014), and on the other hand higher ones, such as ~1:2100 for fluvial changes after a flood event (Tamminga et al., 2015). Although it is not possible to generalize a precision for all settings, ~1:1000 seems an encouraging reference (RMSE of 10 cm for 100 m camera-object distance) to consider for unexplored conditions.

Every combination of field site, camera sensor, ground control points (GCPs) and SfM package in principle requires different photo overlaps, resolutions and perspectives (image network geometry) to achieve certain model accuracy and resolution through UAV-SfM (Elter et al., 2016). This is caused by different surface textures (Cook, 2017), lighting conditions (Gómez-Gutierrez et al., 2014b), camera characteristics (Prosdocimi et al., 2015), GCP characteristics (Harwin and Lucieer, 2012), and SfM algorithms (Eltner and Schneider, 2015). Knowledge to improve the quality of SfM digital surface models keeps expanding by investigating isolated variables, for example, assessing the influences of number and distribution of GCPs (Clapuyt et al., 2016; James et al., 2017) or optimizing camera calibration procedures to manage without GCPs (Carbonneau and Dietrich, 2017). The flexibility, range, high resolution and accuracy that UAV-SfM proved in other conditions shows promising for analysing bank erosion processes throughout the scale of a middle-size river.

The monitoring of bank erosion processes in the case study herein has two specific challenges for the UAV-SfM technique. First, the bank has steep, vertical and undermined surfaces along the domain. Second, the target area is a straight reach with a large length-to-width ratio. The first aspect may require non-conventional UAV paths and camera angles to be able to adequately capture the bank area. The second matter introduces a challenge to georeference the model with rather aligned GCPs, which may result in false solutions rotated around the GCP axis during the model linear transformation. This could be the case since GCPs are to be placed in the bank surroundings to be captured from the UAV, and this target area consist of an overall linear domain. Therefore, the GCP distribution and the image network geometry particularly have key roles in the UAV-SfM workflow applied to measure bank erosion at the process scale.

## 3 Methodology

We used the flexibility of a multi-rotor UAV platform to capture photographs from different perspectives of a 1200 m long riverbank and through SfM photogrammetry derived several DSMs over one year period. We describe the study location in Sect. 3.1, the UAV paths for photo acquisition in Sect. 3.2, and the SfM imagery processing in Sect. 3.3. In order to assess the capabilities of this survey methodology to measure bank erosion at the process scale we proceeded in four steps. First, we verified the elevation precision against 129 RTK GPS points of several DSMs obtained with diverse number of photographs and camera orientations. In this way, we identified an effective number of images to acquire the bank topography with high accuracy. Second, we compared the chosen DSM with airborne LIDAR points to analyse elevation precision over the whole river reach, differentiating between areas of bare ground, grassland and banks. Third, we verified

the georeferentiation accuracy regarding the model rotation around GCP axis. Fourth, we searched for bank features in SfM-based profiles and analogous ones from ALS, and for signatures of erosion processes along sequential SfM surveys.

For the first step, the analysis of the minimum number of photographs needed to achieve the highest DSM precision, we compared the DSMs with RTK GPS measurements to quantify vertical accuracy. We took 129 points across eight profiles on 18-01-2017 (see Fig. 4) with a Leica GS14 RTK GPS, whose root mean square precisions according to the manufacturer specifications are 8 mm + 0.5 ppm in horizontal and 15 mm + 0.5 ppm in vertical directions. On the same date, we flew the UAV along the bank four times with different camera angles and perspectives. Eight photograph combinations were considered to derive 8 DSMs. Then, the comparisons were done with the elevation differences between the GPS points and the corresponding closest ones of the DSM point clouds (e.g., Westoby et al, 2012; Micheletti et al., 2015). We used CloudCompare software (Girardeau-Montaut, 2017) for these computations.

In the second step, we compared the selected DSM from the previous analysis with a reach-scale survey technique, ALS, to analyse topographic differences over the whole river reach. The ALS was carried out on 17-01-2017 from an airplane at 300 meters above the ground level. The laser scanner, a *Riegl LMS-Q680i,* measured a minimum of 10 points per square metre with an effective pulse rate of 266 kHz. We did not have access to the raw data and used the automatically generated 0.5 m grid. We tested the ALS elevation precision against the 129 RTK GPS points using the vertical component of the closest distance to a local Delaunay triangulation of the ALS grid, due to the different resolutions between both datasets. Then, we computed the distances between the ALS grid points and the corresponding nearest ones of the DSM point cloud. We did both computations with the standard cloud/cloud distance tool of CloudCompare, distinguishing between surfaces of grassland, bare ground and bank.

Third, we analysed the DSM spatial stability with respect to the potential axis of rotation around the GCPs, which in the case study laid over a narrow, elongated and straight domain. The GCPs distributed over the floodplain along the near-bank area defined the linear transformation from an arbitrarily scaled coordinate system to the real-world coordinates. In order to verify that the DSM was stable and the tendency to rotate around co-linear solutions did not affect the accuracy beyond the survey target, we computed a regression line with the GCPs to identify the potential axis of rotation for the DSM domain. Then, we projected onto the perpendicular rotational plane the DSM elevation errors corresponding to the GPS points and computed a second regression line to evaluate if there was a linear tendency that indicated a model rotation.

Fourth, we made profiles across six sections of dissimilar erosion rates to contrast the bank representations of i) the SfM DSM, ii) the triangulated ALS grid, and iii) the RTK GPS points. The profiles were computed with MATLAB using i) the Geometry Processing Toolbox (Jacobson et al., 2017) adapted to slice triangle meshes, ii) a linear interpolation across the triangulated ALS grid, and iii) a projection of the RTK GPS points onto the exact cross-section locations. Then, we identified and analysed a cross section over which sequential SfM-UAV surveys showed different stages of the erosion cycle, since the bank erosion cycle was used as a reference to distinguish between techniques capable of measuring at either the process or the cross-sectional scale.

### 3.1 Study site

The study site is a restored reach of the Meuse River, which used to be a single-thread freely meandering river. The river was canalized to a straight reach of 120 m width, the banks were protected and the water levels regulated to improve navigability. However, several kilometres of banks have been recently restored through the removal of revetments and groynes, following the EU Water Framework Directive 2000/60/EC (http://data.europa.eu/eli/dir/2000/60/oj). This reactivated erosion processes to improve the natural value of the river. Important questions have then arisen regarding bank retreat rates and the new equilibrium of the river width. Monitoring bank evolution is necessary to answer these questions and to identify the need, if so, to intervene and at which locations.

The study site is the left bank of a 1200 m long straight reach (Fig. 2) located between the Sambeek and Grave weirs in southeast Netherlands. Seven years after restoration, this reach presents different bank retreat patterns, with sub-reaches of rather uniform erosion and others with embayments of different lengths. Grassy fields used for grazing cover the riparian zone, followed by crop fields across the floodplain. In the near-bank area there are poplar trees every 100 m, some of which have been dislodged during the erosion progression, which is possible to appreciate in Fig. 2 (left) considering the ~200 m embayment between the foreground tree and the next in the background.

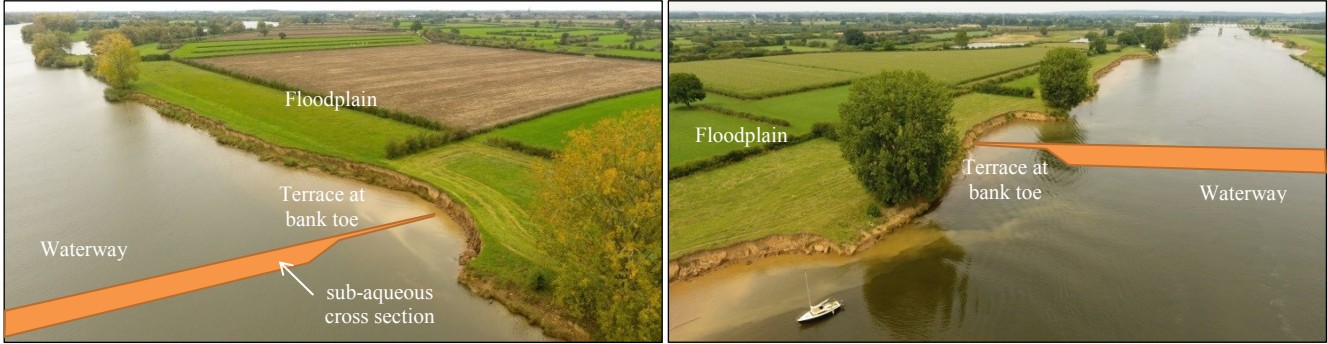

**Figure 2:** Restored bank in the Meuse River, upstream view of first 300 m (left) and downstream view of middle 500 m (right). Bank erosion has caused a series of bays in the 1,200 meter restored reach. Note eroded bank sediment in suspension.

The study site was surveyed eight times with the UAV in 2017. An extraordinary low water level in January provided the opportunity to compare the SfM photogrammetry with ALS and RTK GPS not only for the banks and floodplain, but also for the sub-aqueous terrace at the bank toe (see schematic cross sections in Fig. 2). This terrace was composed of bare soil, without vegetation or obstructions, which adds an extra surface for the comparative analysis. This extraordinary exposure was the consequence of a ship accident against the downstream weir of Grave (on 30 December 2016).

## 3.2 UAV flights for image acquisition

We used the low-cost UAV DJI Phantom 4 to take images of the banks. It has a built-in camera with a 1/2.3" 12 megapixels sensor and a 94° horizontal angle of view. Prior to the image acquisition, a network of Ground-Control Points (GCPs) was distributed on the floodplain to georeference the DSMs (see Fig. 4). The GCP were spaced every approximately 50 metres

along the reach, roughly following the tortuous bankline and avoiding proximity to trees, to simplify the field work and facilitate the GCP visibility from the UAV paths (see Fig. 3-4). This approach missed GCP locations at different elevations, as for instance at the bank toe, and relied on the cross-sectional GCP distribution for the stability of the DSM georeferentiation (see Sect. 3.3). The GCPs were 40 by 40 cm black ceramic tiles (Fig. 3c) fixed to the ground with a circular reflector (12 cm CD) at its centre for their fast recognition in the photographs (Fig. 3d-e show how a GCP is seen from

tracks 1 and 2). We measured the GCP coordinates using the Leica GS14 RTK GPS unit, which was also deployed for the cross-profiling.

An initial flight plan was designed to photograph the banks from four different perspectives, to later compare the results of diverse combinations and find a convenient photo set to survey the target topography in subsequent campaigns. The UAV flew four times in straight parallel lines along the banks (Fig. 3a,b,f) to simplify the set-up and save flying time,

compared to paths that follow changes in bankline or include paths across the domain. The first track took oblique photos from above the river at a height of 25 metres and an average (oblique) distance to the bank of 40 metres (~25 m from the least retreated bankline). The second track had a top view from 40 metres above the floodplain level along the tree line (Fig. 2–4). The third and fourth tracks followed the same path as the second one in respective upstream and downstream directions, but the camera angle was 50 degrees forward inclined from the horizontal plane (see photo footprints in Fig. 4).

These perspectives were thought to capture the tortuous and complex bank surface (Fig. 2, 3a and 4), including undermined upstream- and downstream-facing scarps, with an average ground resolution of 2.1 cm per pixel. We considered a minimum resolution and accuracy of 1/25 times the bank height of the river as a requirement to detect erosion processes, which resulted in a maximum acceptable accuracy of 14 cm for the maximum bank height of 3.5 m at the case study.

We tested five specific combinations of photographs from the different UAV tracks. *Test 1* corresponds to the photo

set of the first track only, which has the side view with the optimal coverage of the bank. *Test 2* stands for the nadir view alone, which is similar to the viewpoint of classic aerial photography. *Test 3* is a combination of the previous two sets. *Test 4* combines tracks 3 and 4, i.e. both paths from above the bank with the oblique forward perspectives in upstream and downstream direction, which allows views on all parts of the irregular banks. Finally, *test 5* utilizes the four tracks with all photographs (Table 1).

We also used the first oblique track to evaluate the minimum longitudinal photo overlap to efficiently capture the bank relief. The photo overlap along the river is a function of the UAV speed and distance to the bank, for a given maximum photo sampling frequency, which in the case of the deployed UAV is one every 2 seconds. Then, flying at 2 m/s along track 1 resulted in 20 photo overlaps for the most retreated areas and 16 for those zones with least bank retreat. Afterwards in

the processing phase, we successively selected a decreasing number of overlaps by twos that resulted in four DSMs. These were *test 1a* when using all photos from track 1 (which is the same set as the aforementioned test 1), *test 1b* when using half of them, and so forth for *test 1c* and *test 1d* (see Table 1).

**Table 1**. Number of photographs and overlaps for the tests

|  | Test 1a | Test 1b | Test 1c | Test 1d | Test 2 | Test 3 | Test 4 | Test 5 |
|---|---|---|---|---|---|---|---|---|
| Track 1 | 293 | 147 | 73 | 37 |  | 147 |  | 293 |
| Track 2 |  |  |  |  | 232 | 232 |  | 232 |
| Track 3 |  |  |  |  |  |  | 232 | 232 |
| Track 4 |  |  |  |  |  |  | 232 | 232 |
| Min. overlaps | 16 | 8 | 4 | 2 | 7 | 15 | 26 | 49 |
| Max. overlaps | 20 | 10 | 5 | 2 | 7 | 17 | 26 | 53 |

### 3.3 SfM workflow

The principles of SfM photogrammetry are similar to those of digital photogrammetry, but the former does not need specifications on camera positions and lens characteristics to reconstruct 3D structures. The camera extrinsic and intrinsic parameters are automatically estimated via tracking and matching pre-defined features in overlapping photos and an iterative
bundle adjustment procedure, which results in a sparse point cloud (Hartley and Zisserman, 2003; Snavely et al., 2008; Westoby et al., 2012). Afterwards, the (dense) point matching is done at pixel scale to generate a detailed point cloud of the scene that has the final survey resolution. The point cloud can then be georeferenced with GCPs, or alternatively, these can be incorporated for the iterative bundle adjustment as additional matched points, during which the georeferentiation takes place.

The use of GCPs is necessary to reference the model to a geographical coordinate system, to compute erosion rates and processes through sequential surveys. The georeferentiation process involves the Helmert transformation of the point cloud through 7 parameters that adjust its scale, position and rotation in a linear and rigid way (Fonstad et al., 2013). The estimation of these parameters is done through a least-squares regression with the GCPs identified in the UAV images. The propagation of linear errors is thus given by the accuracy with which the GCPs were measured (in this study with the RTK
GPS) and then identified in the photographs. Further in the SfM workflow, GCPs can be used to refine the camera parameters estimated during the bundle adjustment, to reduce the non-linear errors that the estimation of the camera parameters may induce (Carbonneau et al., 2017). Ideally, well distributed, precisely measured, and accurately identified GCPs avoid excessive linear and non-linear errors in the point cloud. Yet, a third type of error given by the automated image matching process cannot be prevented with GCPs. These are local and random errors and represent the classic concept of
precision.

We used Agisoft PhotoScan software to process the imagery. For a successful photo alignment from different UAV tracks (Table 1), the camera yaw, pitch and roll recorded during the UAV flight were necessary inputs. For this step we used three GCPs along the reach, two at the extremes and one at the middle, all close to the bank and easily visible from tracks 1 and 2. These approximate orientations and a priori known ground points helped obtaining a consistent sparse point cloud of

5    the bank along the entire reach. The resulting camera positions and orientations of the photo alignment are visible in Fig. 3a, evidencing the UAV tracks. This figure also shows the DSM textured with colours from the photographs, in which the green area on the left side with white patches corresponds to the floodplain partially covered with snow (see also Fig. 3d-f) and the right brownish area is the terrace at the bank toe, with snow remains as well.

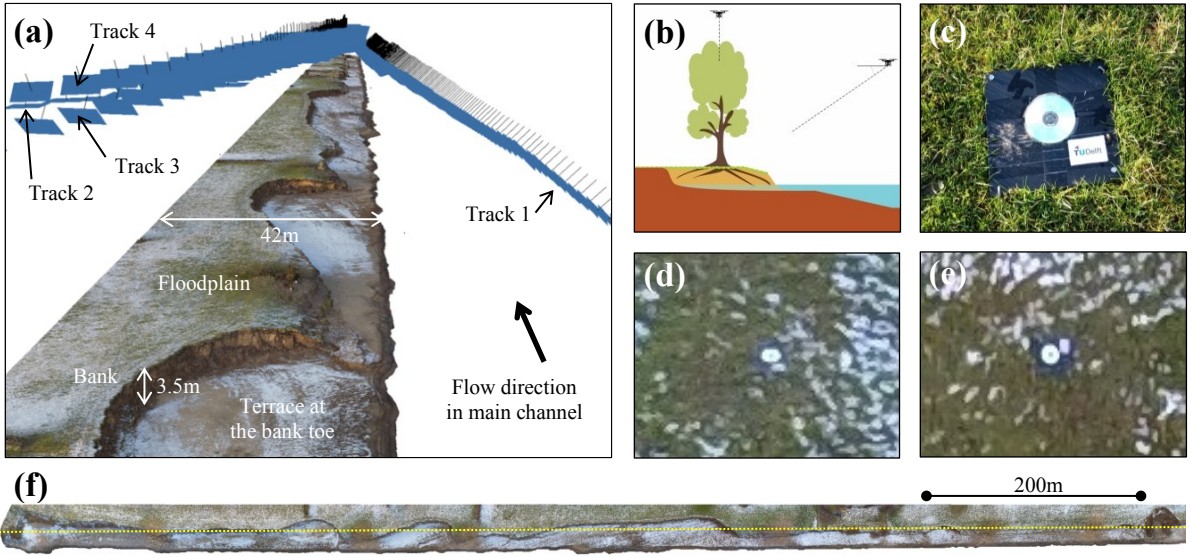

**Figure 3:** a) Camera positions and orientations in perspective view. The digital surface model shows the low-water condition during January 2017, which exposed a terrace at the bank toe. b) Cross-sectional scheme of UAV paths. c) Ceramic plaque with CD as ground control point on the floodplain. d) GCP in photograph from track 1. e) Same GCP from track 2. f) Top view of DSM with UAV track 1 in blue and tracks 2-4 in yellow.

15    After obtaining the sparse point cloud, we marked the remaining 15 GCPs (Fig. 4). Then, we refined the camera parameters by minimizing the sum of GCP reprojection and misalignment errors. This camera optimization adjusts the estimated point cloud by reducing non-linear deformations. Once the dense point cloud was computed, we removed the points outside the area of interest, as well as those points at the water surface, tree canopies and individual bushes at the floodplain. Finally, the point cloud was triangulated and interpolated to generate a triangle mesh. This mesh consisted of a

20    non-monotonic surface that was later processed in MATLAB to plot 2D cross sections.

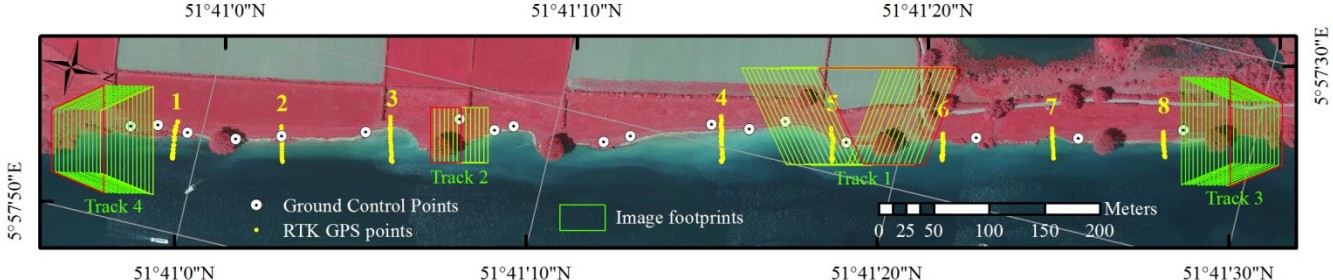

**Figure 4:** Study reach of the Meuse River with GCPs, RTK GPS measurements with cross-section locations and numbers, and some image footprints for all UAV tracks.

## 4 Results

### 4.1 DSM precision: identifying necessary photographs

The sequentially decreasing photo overlaps of Track 1 (Table 1) produced four DSMs, tests *1a–1d*, whose elevation differences with the 129 RTK GPS points are presented in the histograms of Fig. 5. The elevation errors mostly ranged within 10 cm in all tests, but the mean and standard deviation (SD) presented some differences (Table 2 and dot with bar in Fig. 5). Tests *1a*, *1b* and *1c* presented mean values smaller than 1 cm and SD within 3–4 cm, while for test *1d* these values increased to 4 cm and 7 cm respectively (Table 2, rows 1–2). The mean errors on the bank area alone for test *1a*, *1b* and *1c* were lower than 1 cm (Table 2, row 5), but test *1c* had a higher SD of 7 cm compared to 4 and 3 cm of tests *1a* and *1b* respectively (Table 2, row 6). Then, tests *1a* and *1b* had the highest precisions and showed little error differences between them: less than 1 cm for all values in Table 2. Consequently, test *1b* with eight photo overlaps was as effective as test *1a* with 16 overlaps to achieve the highest DSM accuracy. In addition, test *1b* fully covered the tortuous bank area in contrast to test *1c*, especially at the perpendicular stretches of embayments (Fig. 3–4), which assured the choice of 8 image overlaps over 4, despite the general close performance of the latter in terms of accuracy (Table 2, all rows). Therefore, test *1b* became the reference for tests *1* and was used in combination with test *2* to generate test *3*.

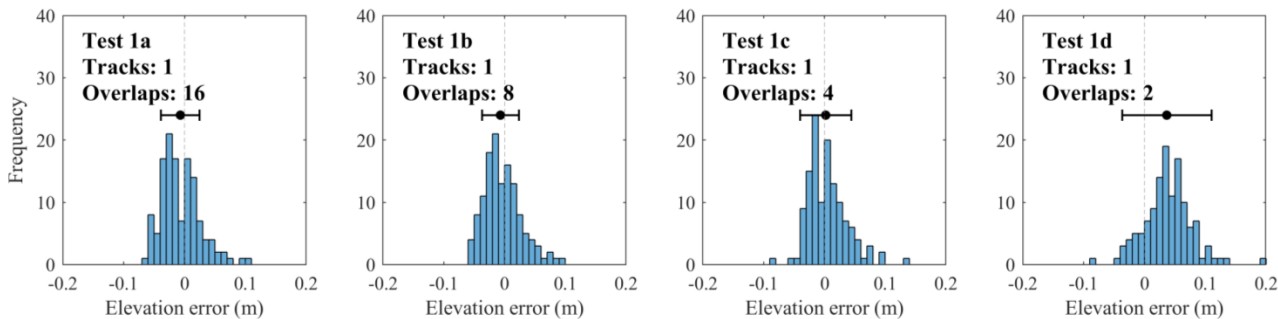

**Figure 5:** Elevation error distributions for SfM tests *1a*, *1b*, *1c*, and *1d*, assuming that the RTK points are correct and without error. Indicated overlaps are the minimum.

**Table 2**. Mean and standard deviation of elevation differences between SfM DSMs and GPS points. Colour intensity indicates the deviation from zero value with minimum/maximum of ±0.13 m.

| Surface | Error (m) | Test 1a | Test 1b | Test 1c | Test 1d | Test 2 | Test 3 | Test 4 | Test 5 |
|---|---|---|---|---|---|---|---|---|---|
| All grounds | Mean | -0.01 | 0.00 | 0.00 | 0.04 | 0.03 | -0.01 | -0.05 | 0.00 |
| | Std. dev. | 0.03 | 0.03 | 0.04 | 0.07 | 0.03 | 0.03 | 0.05 | 0.03 |
| Grassland | Mean | 0.02 | 0.01 | 0.01 | 0.02 | 0.02 | 0.01 | 0.01 | 0.02 |
| | Std. dev. | 0.03 | 0.03 | 0.04 | 0.04 | 0.03 | 0.02 | 0.02 | 0.02 |
| Bank | Mean | 0.00 | 0.01 | -0.01 | -0.01 | 0.05 | 0.01 | -0.03 | 0.01 |
| | Std. dev. | 0.04 | 0.03 | 0.07 | 0.13 | 0.03 | 0.03 | 0.04 | 0.03 |
| Terrace | Mean | -0.02 | -0.02 | 0.00 | 0.06 | 0.03 | -0.02 | -0.09 | -0.01 |
| | Std. dev. | 0.02 | 0.03 | 0.03 | 0.06 | 0.03 | 0.02 | 0.03 | 0.02 |

Figure 6 shows the error distributions of the remaining four DSMs, i.e. tests *2–5*, which also were mostly within 10 cm, except for Test *4*. This test had evident higher errors than the rest, mostly concentrated at the terrace (Table 2, row 7). Tests *3* and *5* had the lowest mean elevation errors, both lower than 1 cm, with the same SD at all surfaces that were lower than 3 cm. Test *2* presented a similar SD, but the mean was biased 3 cm. This test in combination with test *1b* slightly reduced the SD errors of the latter (Table 2, rows 7 and 6), but without significant overall improvements. All in all, tests *1b*, *3* and *5* had the best performances with average errors lower than 1 cm and standard deviations within 3 cm, however with increasing number of photographs (Table 1). The most efficient one was then test *1b* that used the lowest number of photographs to achieve similar precision, especially on banks.

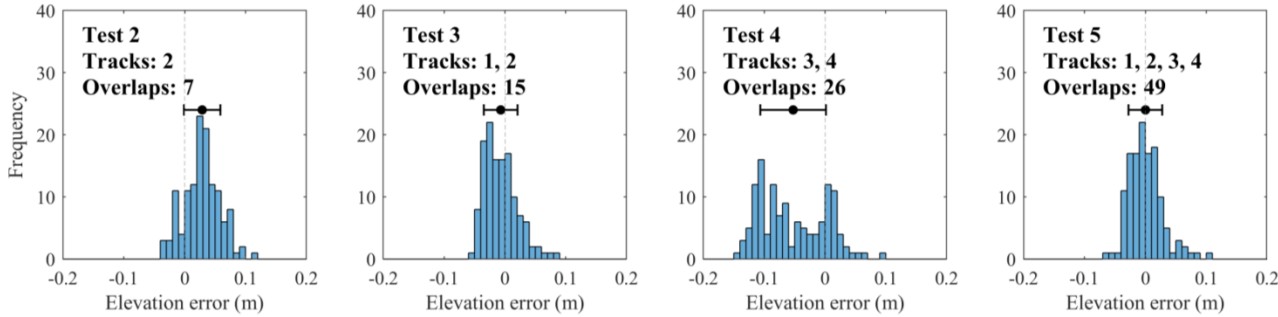

**Figure 6:** Elevation error distribution for tests 2, 3, 4, and 5. Ordinates indicate number of GPS points in each bin.

Interestingly, if we consider all tests, the elevation errors on grassland were similar to each other (Table 2, rows 3 and 6), means between 1 and 2 cm and SD between 2 and 4 cm, whereas the bank and terrace did not present this behaviour. Furthermore, while the bank values (Table 2, rows 4 and 7) did not correlate with those of all grounds (Table 2, rows 1–2), the terrace mean elevation differences (Table 2, row 5) linearly correlated with those of all grounds (Table 2, row 1) with $R^2$ = 0.97. Therefore, the error biases for all grounds throughout the tests were most likely due to the biases from the points over the terrace.

To conclude, despite virtually doubling the number of images in comparison with test 1b, the test *3* setup with a nadir track and a side-looking track was chosen for subsequent UAV surveys on the basis of two findings. First and most important, growing vegetation at the bank toe occluded parts of the target surface from the oblique camera perspective. Second, the GCPs on the floodplain laid almost horizontal, which made them easier to identify from the top-view during an initial phase of GCP recognition in the photographs. Moreover, we found at later surveys that growing grass on the floodplain was sometimes blocking GCP plaques from the angle of vision of UAV track 1, for which using the nadir view of track 2 was advantageous to locate the plaque centres, preventing the otherwise disuse of some GCPs.

**4.2 DSM precision over the reach: comparison with ALS**

Compared to the ALS grid, test *3* point cloud showed a good agreement over most of the reach. This is observable from Fig. 7a, corresponding to the blue areas that indicate elevation differences lower than 5 cm. Yet, two notable regions surpassed this difference: the bank and the extremes of the reach. The latter were zones beyond the GCPs, where higher errors in the DSM are expected when using parallel image directions due to inaccurate correction of radial lens distortion (James and Robson, 2014; Smith et al., 2014). Consequently, the results outside the GCP limits cannot be considered representative of the whole domain and they were discarded for the subsequent statistical comparisons (beyond the dashed lines in Fig. 7). Within the GCP bounds, the bank area presented relatively high elevation differences, which makes the bankline visible in Fig. 7. Moreover, another sloped area at end of the terrace also presented higher differences than surrounding areas, which is visible as a thin light-blue line at bottom of the domain in Fig. 7a (see also Fig. 3a-b for other perspectives of this slope toward the channel bed).

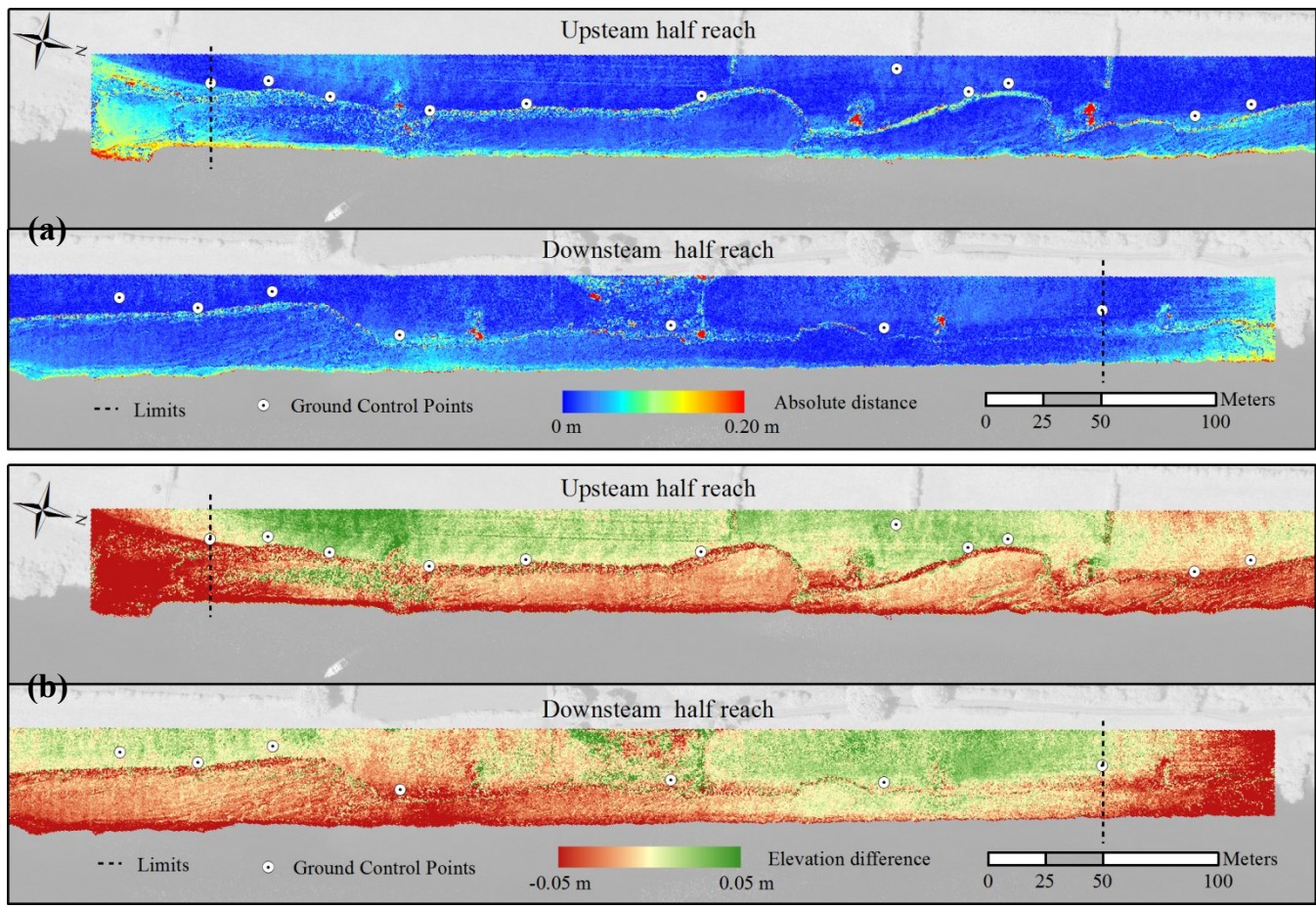

**Figure 7:** a) Absolute elevation differences between SfM and ALS along the reach. b) Signed elevation differences between SfM and ALS with a smaller scale range.

Figure 7b presents signed elevation differences with a smaller scale range to highlight areas with positive and negative deviations between techniques. Green zones indicate higher ground elevations on the SfM DSM than in the ALS grid, red indicate the opposite, and yellow elevation matches. The upstream half domain presented a general tendency of SfM to overestimate elevations on the floodplain, and in turn underestimate them on the terrace. This trend is also observed in the upper part of the downstream reach. Yet, these zones showed exceptions, such as a green patch at the terrace close to the dashed limit, and a red patch on the floodplain at the end of the upstream half reach. Despite the described general opposed behaviour between floodplain and terrace, the downstream reach evidenced two zones with consistent trends across the domain, covering both the floodplain and terrace. First, lower SfM elevations at the end of the largest embayment (downstream half reach), and second, SfM higher elevations at the end of the reach, before the dashed limit.

Figure 8 presents the relative frequency distributions of the elevation differences divided into three regions: the grassy floodplain, the steep bank, and the bare-ground terrace. Over the grassland, both SfM and ALS had rather similar results (Fig. 8, centre left), with 1 cm mean difference and 2 cm of standard deviation (Table 3). In contrast, the bank had a

bias between techniques of 6 cm (Table 3) and a relatively high standard deviation of the same value. Finally, the terrace showed a slightly higher deviation than over the grassland (Fig. 8 and Table 3) but with a bias of -4 cm. The bank area together with the terrace induce an overall small negative bias in the elevation difference distribution (Fig. 8 left and Table 3). The former with a small contribution to the total number of measurements and the latter with a greater number but a lower magnitude.

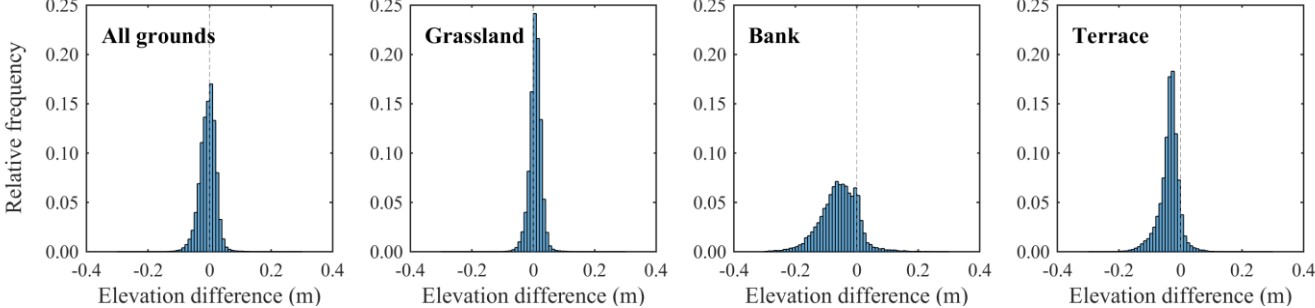

**Figure 8:** Comparison of elevation differences between SfM and ALS for distinct surface types

Table 3 also indicates the differences of SfM DSM and the ALS with the 129 RTK GPS points. Interestingly, the ALS presented a constant bias of 1 cm across all surfaces, but the standard deviation did change significantly among them: the bank had a standard deviation of 9 cm, which doubled the deviation of the terrace and tripled that of the grassland. While the SFM DSM had comparable absolute biases than those of ALS, the standard deviations were all respectively lower. Particularly at banks, the standard deviation of the SfM DSM was only 3 cm in contrast to the 9 cm of the ALS, which makes the former approach considerably more accurate than the latter. This could explain the relatively large elevation differences between the two methods in the bank area (Fig. 8, centre right), occurring due to a lower precision of the ALS and not vice versa.

**Table 3**. Mean and standard deviation of elevation differences between SfM, ALS and RTK GPS. Colour intensity indicates the deviation from zero value with minimum/maximum of ±0.09 m.

| Subtraction | | All grounds | Grassland | Bank | Terrace |
|---|---|---|---|---|---|
| SfM - ALS (m) | Mean | -0.01 | 0.01 | -0.06 | -0.04 |
| | Std. dev. | 0.03 | 0.02 | 0.06 | 0.03 |
| ALS - GPS (m) | Mean | 0.01 | 0.01 | 0.01 | 0.01 |
| | Std. dev. | 0.05 | 0.03 | 0.09 | 0.05 |
| SfM - GPS (m) | Mean | -0.01 | 0.01 | 0.01 | -0.02 |
| | Std. dev. | 0.03 | 0.02 | 0.03 | 0.02 |

### 4.3 DSM rotation around GCP axis

A regression line was computed with the locations of the GCPs to analyse with respect to this axis the rotational tendency of the model. The GCP distribution around the regression line and the bankline are shown in Fig. 9 (left). The adopted GCPs spanned 19.7 m across the reach and 1.6 m in the vertical direction (Fig. 9, right). The bank scarp along the reach, which is the target survey area, covered 26.9 m in cross-sectional direction due to the wide-ranging erosion magnitudes of the case study, with maximum bank heights of 3.5 m, indicated with a grey area in the right panel of Fig. 9. This side perspective of the domain evidences the potential plane of rotation, whose stability depends on the position of the GCPs.

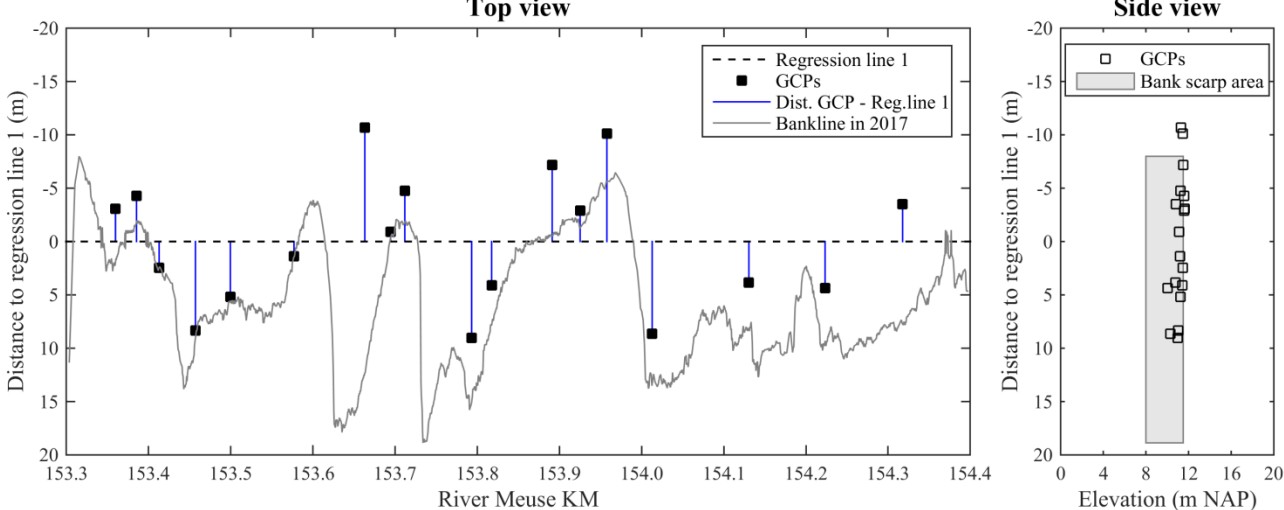

**Figure 9:** GCP horizontal distances to regression line (left) and respective positions in potential rotation plane from the regression line (right)

The rotation potential of the model is evaluated comparing SfM DSM elevations with those of the 129 GPS points (Fig. 4). Fig. 10a presents the locations of the GPS points used for accuracy control projected at the potential rotation plane, showing that they covered the domain across the channel (abscissa axis) and different elevations along of the scarp area (ordinate axis), resulting in a reasonable sample to assess the model georeferentiation stability. Fig. 10b presents the DSM elevation errors distributed across the regression line 1. In this plane, a second regression line was computed with all 129 points, represented with a dashed black line in Fig. 10b. This line is tilted from the horizontal suggesting that the model was rotated with a magnitude equal to the respective slope.

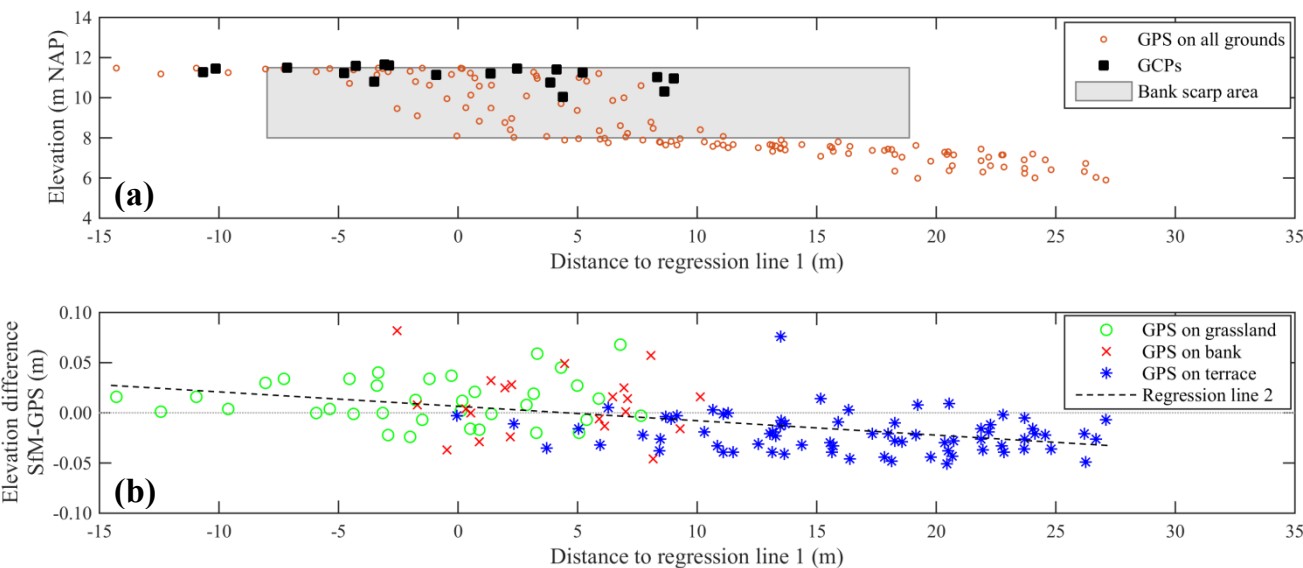

**Figure 10:** a) Location of GPS measurements in potential rotation plane around regression line 1. b) SfM DSM elevation errors across regression line 1, distinguished among areas of grassland, bank and terrace.

The GPS points corresponded to difference surface types, shown with different point markers in Fig. 10b. It is clear that on the left side to the regression line 2 down-crossing, errors mostly had a positive bias, whereas on the right they mostly presented a negative bias. At the same time, those errors with positive bias were generally over a ground covered with grass, and those with negative trend were measured on a slightly sloped bare ground (see also Table 2, column for Test 3). Then, this tendencies could also be ascribed to the overestimation of the grass cover in the former case, and a non-linear transverse deformation beyond the GCP bounds, for the latter. Regardless of the causes, the results showed an overall transverse DSM inclination with respect of the GPS points used for accuracy control. The regression line 2 when evaluated at the extremes of the bank scarp area (-7.99 m and +18.87 m) yielded an elevation difference between these points of 3.9 cm.

## 4.4 Bank erosion features and process identification

Six bank profiles were selected among those surveyed with GPS on January 2017 (Fig. 4) to compare the bank representation with the different survey techniques. Fig. 11 shows the bank profiles at sections 1, 2, 4, 6, 7 and 8. These sections presented distinct erosion magnitudes and features after seven years of restoration, for example, section 8 (Fig. 4 and 11) appeared close to the original condition, with a mild slope and nearly no erosion, whereas sections 6 and 7 had vertical scarps. The SfM DSM profiles are represented by continuous lines, the ALS profiles with dashed lines, and GPS points with circles. The SfM representation had better proximity to the GPS points than the ALS in almost all cases. What is

more, ALS generally overestimated the elevation corresponding to the GPS points, which confirms the bias observed in the comparison of bank elevations shown in Fig. 7 and 8.

SfM profiles showed detailed bank features, such as a collapsed upper bank laying at the toe (section 2), an overhang at the bank top (section 1), small-scale roughness on scarps (sections 6 and 7), and slump-block deposits (section 4). These features appeared as simple shapes in the profiles but they were confirmed with field observations. The ALS depicted simpler profiles, smoothed by coarser resolution, which made it difficult to identify characteristic features of the erosion cycle in them. Yet, ALS profiles had enough point spacing to capture gentle bank slopes with reasonable precision (section 8), but for steeper ones (sections 1, 2 and 4) and specially at scarps (sections 6 and 7), this technique provides lower accuracies.

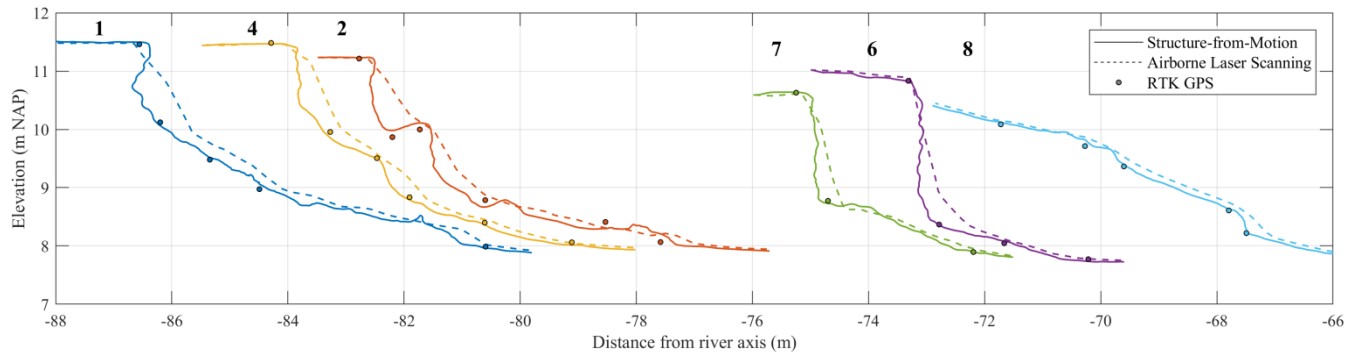

**Figure 11:** Banks measured with SfM (continuous lines), ALS (dashed lines) and GPS (circles) on 17−18 January 2017. Cross-sections are located from left to right, at river km. 153.4, 153.9, 153.5, 154.2, 154.1, and 154.3 (see Fig. 4 for locations).

The temporal development of section 4 (Fig. 4 and 11) is illustrated in Fig. 12a by a sequence of SfM-UAV surveys. The initial stage corresponds to the survey of Fig. 11 on 18 January 2017. The consecutive surveys showed the evolution of the vertical bank profile, through which different processes can be inferred. The bank profile, initially characterized by a top short scarp and slump blocks along the bank face, experienced a mass failure and a further removal of blocks between January 18 and March 15 2017. Between March 15 and April 26, only toe erosion occurred. By June 21, another mass failure happened, which left slump blocks along the lower half of the bank. On July 19, these blocks were removed, leaving a steep bank face. Then, further toe erosion caused a small soil failure at the lower bank whose remains laid at the toe. On October 11, this wasted material was removed. Then, until the last survey on November 22, entrainment occurred at the lower half of the bank profile, further steepening the bank. In light of the results, the methodology resolution and accuracy are high enough to identify different phases of the erosion cycle, enabling the analysis of bank erosion processes in conjunction with data on potential drivers.

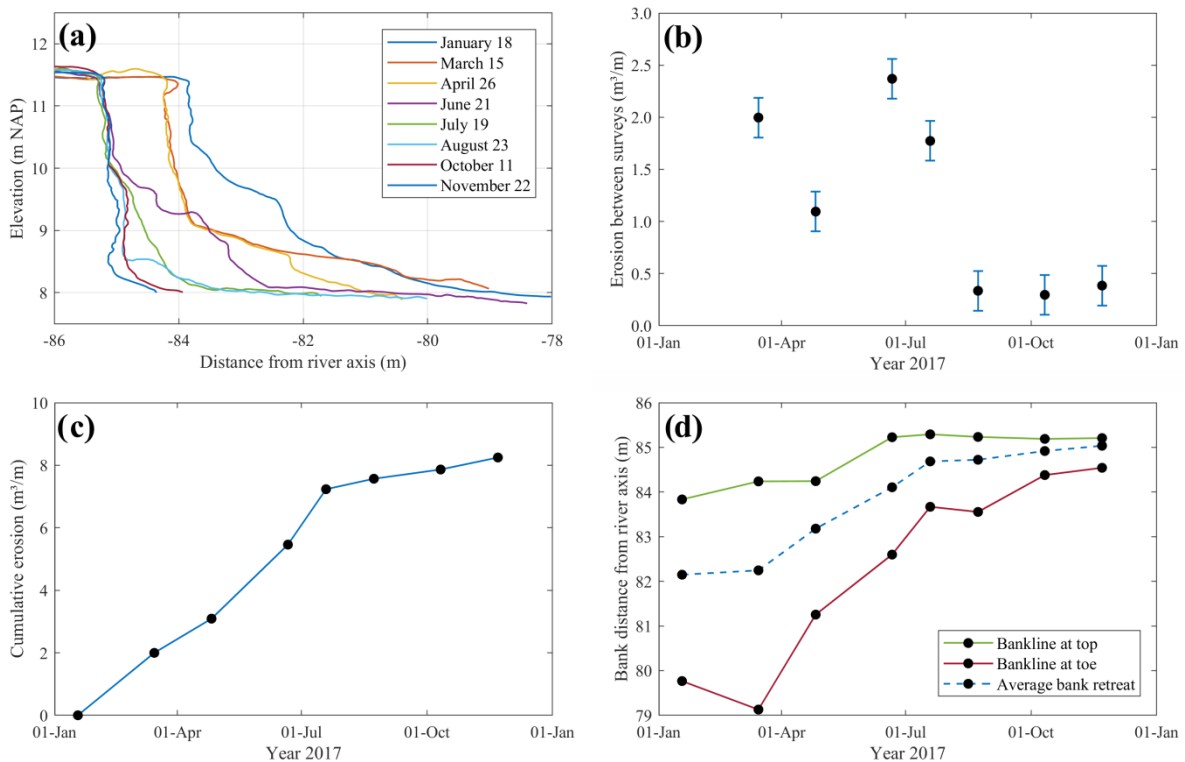

**Figure 12:** Sequential surveys at cross section 4, Meuse River km. 153.9, over 2017. a) Bank profiles from DSMs; b) Eroded volume per unit width between consecutive surveys; c) Cumulative erosion along surveys; d) Bankline locations at the top and toe of the bank.

In addition to process description, the quantification of eroded volumes is possible computing the net area between sequential bank profiles. For example, Fig. 12b shows eroded volumes per unit width between consecutive surveys plotted at the end of each time interval, with an error bar based on the RMSE of test 3. Evidently, there were different erosion rates during the year and the highest ones happened in the first part of it. Fig. 12c presents the respective cumulative eroded volumes per unit width, where the two trends can be distinguished: a gentle slope towards the end and higher rates of sediment yield during the first haft of the year. Given that the topographic measurements are limited to a single year, it is not possible to state whether this behaviour in recurrent on a yearly basis. However, this case exemplifies the possibilities to quantify eroded volumes throughout different phases of the erosion cycle.

The bankline retreat as a measure of bank erosion involves the identification over time of the bank top, but this concept could be extended, for instance, to the bank toe. Fig. 12d shows the temporal progression of the bankline distance from the river axis for both the bank top and the toe, which we arbitrarily defined for this case at 11.1 m and 8.1 m, respectively. The top bankline showed a mayor jump between April and June and a smaller one between the first two surveys, corresponding to mass failure events. The bank toe presented a more gradual retreat, with events of slumping and temporal accretion that were timely captured along the surveys. This alternative bank retreat representation provides

evidence of the development of bank erosion at every survey. The contrast of bankline retreats at the top and toe of the bank illustrates how different processes on their own represent dissimilar erosion evolutions, since they constitute different phases of the erosion cycle, i.e. at the top mass failures and the toe slump block removal and entrainment. Finally, the average bankline between bank toe and top would best represent the real retreat (dashed line in Fig. 12d), despite not necessarily indicating an actual bank location for a specific elevation. This approach logically considers all erosion phases and follows a similar trend as the cumulative erosion of Fig. 12c.

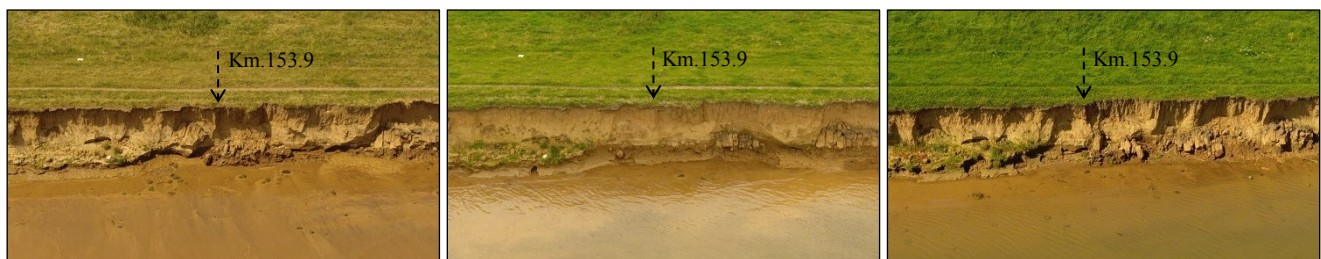

**Figure 13:** Zoomed-in UAV photos from track 1 at section 4, km. 153.9, showing diverse bank erosion stages: slump blocks at bank toe on June 21 (left), clean bank face after block removal on July 19 (centre), and undermining on August 23 (right).

Finally, the availability of the UAV imagery provides additional information to analyse and interpret bank evolution through direct observation. Figure 13 shows photographs from the three consecutive surveys on June 21, July 19 and August 23 at section 4, km. 153.9. At this bank area, the sequence shows on the left side of the panels only entrainment at the bank toe, and from the centre (km. 153.9) to the right, one or more erosion cycles. Cross section 4 presented on June 21 (Fig.13, left panel) slump blocks laying at the bank toe that were removed by July 19. Then on August 23, this cross section had incipient undermining at the lower bank and block deposition at the toe. The bank area at the right of section 4 experienced a second mass failure after June 21, completing one erosion cycle, and further downstream it is more difficult to keep track of the cycles due to faster erosion rates. The UAV photos also evidence the progressive growth of grass over the floodplain, especially observable next to the walking path, which was captured along the surveys (Fig. 12a).

## 5 Discussion

### 5.1 UAV flight and SfM precision

In general, there were no large differences in accuracy between the DSMs derived with different photo perspectives and overlaps. The accuracy of the tests, except for test *4*, was approximately 10 cm complying with the target accuracy and resolution of 14 cm, and they all represented characteristic features of the erosion cycle, such as slump block deposited at the bank toe and mass failures. Yet, other topographic features that were hidden from the nadir UAV perspective, such as undermining, were only captured from oblique camera perspectives. For instance, the area below the top overhangs visible at cross sections 1 and 2 (Fig. 11) were not captured in test *2,* and were represented with a lower resolution in test *4*. The UAV

viewpoint of track 1 not only had the largest bank area coverage compared to the other camera perspectives proposed in this work, but also achieved the highest elevation precision without the need of other tracks. Yet, the nadir view of track 2 contributed to cover an additional bank area behind trees and bushes growing at the bank toe along the first 200 m of the reach (Fig. 2, left), for which it was complementarily used with track 1. Since vegetation can occlude the bank face, if denser

and more abundant, it could prevent the usage of the survey technique, in a similar way as high water levels do.

The results herein show that, in the absence of bank toe vegetation, a single oblique UAV track with eight photo overlaps and visible GCPs appears effective to survey banks with the highest precision and coverage, for the given sensor size and resolution, camera-object distance and lighting conditions. This number of photo overlaps agrees with the laboratory experiment of Micheletti et al. (2015), who found that above eight the mean error was only slightly decreased, in contrast to

increasing overlaps within the range below eight. Nevertheless, they showed that overlaps higher than eight reduced the number of outliers, a trend which in our case is evident for less overlaps: test *1c* (4 overlaps) mainly differed from test *1b* (8 overlaps) in a higher RMSE but not in the mean. This difference may arise from the distinct texture and complexity of each surface, which presumably requires different number of images for a similar performance (James and Robson, 2012; Westoby et al., 2012; Micheletti et al., 2015).

A RMSE of 2.8 cm to measure a riverbank with the photo combination of test *3* results in a relative precision with respect to the average camera-surface distance of 0.0007 or ~1:1400. This relative precision ratio is somewhat higher than ~1:1000 achieved by James and Robson (2012) for steep irregular features at kilometre-scale in a volcanic crater and decametre-scale in a coastal cliff, whereas our precision is somewhat lower than ~1:2000 those authors proved at metre-scale. More precise results could be possible using a bigger and higher-resolution sensor, flying closer to the bank, or even

trying other oblique bank perspectives. However, this endeavour would only be reasonable if such data are needed for research purposes and if GCP positioning had also according higher precisions, since registration errors translate into the DSM accuracy during camera parameter optimization and/or georeferentiation (Harwin and Lucieer, 2012; Javernick et al., 2014; Smith et al., 2014).

A precision of 10 cm has implications for the representation of small-scale features at bank scarps. Despite the

presence of features in the order of decimetres, we could not assess their accuracy given the discrete GPS points and the 0.5 m ALS grid used to assess the DSM precision. For instance, Fig. 12a shows an upper bank scarp along the last four surveys that, if assumed unchanged, it would indicate a maximum distance of 20 cm between surveys, which still would remain within the ±10cm error estimated by the GPS comparison. Although these differences could have been caused by weathering processes or growing grass on the bank face, potential sources of error at such scale could be given, for instance,

by registration errors or occlusions caused by the surface roughness (Lague et al., 2013). Then, further research is needed to evaluate the precision at the roughness scale to, for example, analyse form drag at the bank face (Leyland et al., 2015).

The analysis made for test 3 on model rotation evidenced a linear trend with increased surface elevations on the floodplain side of the domain and decreased elevations on the main channel side (Fig. 10b). This tendency was probably caused by a rotation of the DSM around somewhat aligned GCPs that may lead to co-linear solutions of the Helmert

transformation (Carbonneau and Dietrich, 2017). Yet, the areal SfM-ALS comparison within ± 5 cm range (Fig. 7b), did not evidence a clear axis along the whole domain that could suggest a rigid rotation of the DSM, since the ALS did have a constant mean accuracy across all surface types (Table 3). The linear rotation tendency, then, might have been obscured by the error range that was larger across the domain (Fig. 6 centre left and 10b) than the mean rotation magnitude, whose

elevation difference between the extremes of the cross-sectional domain was 6 cm, and 4 cm within the bank area. Thus, other sources of error were also present that resulted in the obtained error range.

The comparative analysis of the DSM elevation errors from different photo combinations showed that the ground surfaces surveyed in the case study had different precisions. The grassland presented similar errors with a positive bias throughout all tests. The positive elevation differences are typical of vegetated surfaces (Westoby et al., 2012; Micheletti et

al., 2015), whereas the similar performance of different photo combinations might be due to the presence of sufficient and well distributed GCPs in this area (the floodplain). The terrace at the toe of the bank, in contrast, presented different error skewness throughout the tests, which affected the error distribution for all grounds. Interestingly, the error deviation of tests 1 increased as the overlaps decreased, which in turn implies that more overlaps created more robust models. The linear errors cannot explain this behaviour because the same GCP locations were used for all tests, and only the camera parameters

were optimized for each.

The error skewness at the terrace throughout the tests could be related to the fact that this area was the most distant from the GCPs and it was not surrounded by them, so that errors in lens distortion corrections could have especially increased here (James and Robson, 2014; Javernick et al., 2014; Smith et al., 2014). This effect was clear at the reach extremes (Fig. 7), where the elevation differences increased with respect to the ALS survey further from the GCPs, for which

it is called 'dome' effect. While James and Robson (2014) showed that using different (convergent) camera angles is effective to mitigate the 'dome' effect, our results showed that the DSM precision with eight photo overlaps along a single UAV track did not substantially improve by adding the extra perspective of track 2. This may imply that the chosen number of overlaps and used GCPs were sufficient to avoid distortions in the bank area that exceed the required accuracy, together with the fact the track had oblique and not nadiral perspective.

It is most likely that all the mentioned types of error were present in the SfM DSM, i.e., linear errors given by rotation (linear trend of errors across potential rotation plane, Fig. 10b), non-linear errors given by the estimation of camera parameters (patches of higher or lower SfM elevations across the domain compared to ALS, Fig. 7b), overestimation of ground elevation with grass cover (Table 2, Fig. 7b, Fig 12a), and random errors given by the bundle adjustment that could not be assessed herein. Although, the adopted workflow was effective to measure bank erosion processes with the target

accuracy, linear and non-linear errors could have been reduced, for instance, using a larger cross-sectional GCP distribution or better visible and bigger GCP targets. Other possibilities are also open, such as combining two oblique perspectives with a second angle better capturing the floodplain, GCPs and bank area.

## 5.2 Comparison of two reach-scale techniques: SfM and ALS

The elevation bias at the bank between the SfM-based DSM and the ALS grid (Fig. 8) was caused by the topographic overestimation of ALS (Table 3 and Fig. 9). This ubiquitous error is ascribed to a known limitation of ALS systems related to the laser beam divergence angle, which locates the closest feature within the laser footprint at the centre of the footprint. This increases the ground elevation at high-slope areas (Bailly et al., 2012), which is the case for riverbanks. Still, the ALS resolution and precision were enough to identify bank slopes, in accordance with other studies (e.g., Tarolli et al., 2012; Ortuño et al., 2017). Furthermore, despite the ALS capability to estimate volume changes of eroded banks (Kessler et al., 2013), the method omits information related to the phases of the erosion cycle by not surveying erosion features smaller than its resolution (in this case 0.5 m), apparent in contrast with SfM profiles (Fig. 11). Moreover, if finer ALS resolutions are available, for instance using higher frequency lasers or working with the raw data, more ground details can be captured, but still vertical or undermined profiles would be missed.

The elevation differences between the methods observed for grassland (Fig. 8, centre left) were probably caused by dissimilar ground resolutions, because a larger elevation scatter is expected in the SfM-based DSM when capturing grass with 2 cm resolution, compared to the interpolated ALS samples into a 50 cm grid, even when derived from 0.16 m beam footprints. Nonetheless, the mean difference was zero (Table 3), so that both methods overestimate in the same way the real ground elevation due to grass cover. The effect of this is visible, for instance, in the increasing surface elevations on the floodplain over a year (Fig. 12a), which happens after the mowing period in October. The terrace at the bank toe presented a similar scatter as grassland, but had a small negative bias that could be explained by a DSM rotation or a slight transverse 'dome' effect of the SfM DSM.

The distance covered by the SfM-UAV method depends on the flight autonomy. The deployed UAV had autonomy of approximately 25 minutes, which limited the maximum bank survey extent to approximately 2 km for the tested UAV height and speed, and camera resolution and shutter frequency. This practical limit will change with the progressive development of UAVs, but the distance covered by a single flight is currently significantly smaller than the one covered by ALS. Although a larger camera-object distance and speed than the used in this work would increase the surveyed area, decreasing the ground resolution and the UAV stability may result in the loss of sufficient detail to capture erosion features, and what is more, decrease the DTM precision that depends on the image scale (James and Robson, 2012; Micheletti et al., 2015). Therefore, further investigations would be required to explore the practical limits of UAV-bank monitoring in views of extending the survey coverage.

## 5.3 UAV-SfM challenges to measure bank erosion processes

The use of UAV-SfM to measure bank erosion processes presents specific challenges, since bank areas usually have vertical surfaces and lengths can be much larger than the other two dimensions. Furthermore, the reach under analysis was particularly straight, introducing additional complexity to apply the technique.

### 5.3.1 Vertical surfaces

The bank presented very steep, vertical and undermined surfaces that required an oblique camera perspective to adequately capture it. On the other hand, a camera from a top view omitted undermined areas and reduced the DSM resolution at the bank. It was then necessary to include an oblique UAV camera in the flight plan to measure bank erosion at the process scale, but in turn this required that GCPs needed sufficient visibility from this angle. In addition, it was convenient to have accessible GCP in the field for their placement and sometimes later removal, so their location across the bank became important too, from a practical point of view.

The accessibility and visualization of GCPs depend on where and how they are placed across the bank profile. In our adopted approach, GCP targets were horizontally placed over the floodplain and close to the bankline, covering 20 m across the channel, which was convenient for a fast field campaign. Yet, GCPs were not placed at the bank toe or over the bank face, so a very limited vertical extent was covered (Fig. 9-10), and at the same time, the target area was not surrounded by GCPs along the three dimensions. Then, the rotation stability of the DSM and the non-linear effects beyond GCP bounds relied on the horizontal, and particularly the cross-sectional, GCP distribution, as well as on the number of photo overlaps that reduced the error scatter (see Sect. 5.1). Although a larger vertical range, as with GCP at the bank toe, may be effective to reduce potential non-linear errors at the bank, the linear errors may not significantly be reduced in this way given the relatively short bank height (~3.5 m) and corresponding horizontal extent. In contrast, the relatively large cross-sectional GCP span possible on the floodplain (e.g., 20 m) is clearly more effective to avoid model rotations around the GCP axis (see Fig. 9). Finally, in our case study, there were no clear non-linear effects at the bank area that could justify the placement of GCPs along the bank face.

The visible dimension of the GCP is proportional to the cosine of the viewing angle with respect to the normal of the plane in which the GCP lies (say $\alpha$), which increased the uncertainty to locate the target centre along the transverse direction, thus hindering the model georeferentiation when a single UAV track was used. On the other hand, targets were not always distorted in longitudinal direction from oblique angles, i.e., when the camera was near the GCP cross-section, so horizontal errors were not as sensitive along the river axis as they were across the channel. Linear positioning errors in the transverse direction directly affect the accuracy to quantify erosion rates since this is based on the change of bank face positions over time. Moreover, the error introduced by a coarser resolution translated into elevation errors, and these directly affect rotational errors. On the other hand, the lateral view helped to compensate for this, since this elevation errors decreased with the cosine of $\alpha$. Although the proposed approach attained the required DSM accuracy, the mentioned shortcoming could been solved by orienting the GCP plaques towards the oblique camera, placing them somewhat parallel to the bank surface. This could have been done, for instance, with a back stand on the rigid plaques.

### 5.3.2 Linear domain

Banks, considered from the reach scale, are linear domains that extend along the river with the other two dimensions much shorter than their length. The application of UAV-SfM to these particularly elongated domains run the risk of having rotated solutions of the Helmert transformation when georeferencing the DSM with GCPs. The proposed workflow intended to avoid linear rotations of the DSM through sufficiently accurate and well-distributed GCPs, while using parallel UAV tracks along the river, which did not contribute to the model stability. As discussed in Sect. 5.1, the cross-sectional GCP distribution and the accuracy of GCPs were essential to achieve the desired DSM accuracy by reducing linear and also non-linear errors, which depended on both the field measurement and the identification in the photographs.

The choice of UAV tracks parallel to the river axis simplifies the mission set up and shortens the flying time too. Yet, this configuration does not provide additional stabilization to the DSM because it is closely aligned with the potential axis of rotation. For instance, the combination of flight tracks that covered a wide cross-sectional distance could contribute to reduce the model rotation through the locations given by the UAV internal GPS (Carbonneau and Dietrich, 2017). Nonetheless, such approach would be limited by relatively high GPS errors for the accuracy needed to measure bank erosion processes. Thus, this method would require large distances between tracks across the floodplain to significantly reduce the influence of those positioning errors on the model rotation. The adopted approach relied on GCPs to georegister the DSM, which proved effective despite using parallel-axes image orientations. What is more, since parallel UAV tracks tend to increase doming effects (James and Robson, 2014), GCPs were also important to prevent excessive non-linear errors for the required accuracy. Fig. 7b showed no significant DSM deformations within the GCPs, but increasing errors outside the GCP bounds, for which they need to be carefully distributed.

In principle, the cross-sectional GCP distribution should be as wide as possible to ensure that the model georeferentiation is stable. Then, a set of GCPs should be placed close to the bankline, which is also beneficial to reduce non-linear errors at the bank area and to visualize them from the oblique camera, and another set far away from the bankline. This second group, on the other hand, would have lower resolutions in the UAV photos due to larger distances to the camera, and higher related elevation errors during the target identification on the images. A compromise is then necessary between a minimum GCP cross-sectional span and the target resolution, especially for a flight plan with a single oblique UAV path. In our case, a GCP distribution spanning 20 m across the floodplain satisfied the required DSM accuracy of 1/25 the bank height, but improvements are possible in this respect.

For instance, if the first GCP line is 40 m from the UAV camera in oblique direction (Fig. 3b), the second line could be placed 51 m inland from the first line to be 80 m from the UAV sensor (twice as far), but with targets four times as big as those next to the bankline to linearly compensate for the decrease in the image resolution (although no linear trend was found between errors and sensor-object distance, Eltner et al., 2016). In this example, the rotational error would decrease 2.5 times compared to the results presented here, given by the increase in the cross-sectional GCP footprint and considering similar elevation errors (from GPS and image identification). Finally, as indicated in the previous section, it is advisable to use

inclined targets perpendicular to the line of sight of the camera, which could have double spacing compared to those in the front while keeping the number of photo overlaps that capture them. This method would require more time in the field to place and measure GCPs but certainly would achieve higher DSM accuracies from lower linear errors.

### 5.3.3 Other considerations

A single oblique path was sufficient to achieve the highest accuracy among the different tests (Sect. 4.1), but using only this track when there is no bank-toe vegetation would still present a challenge to georeference the DSM, as already discussed. In this work, we opted to use both oblique and nadiral perspectives because of the presence of bank-toe vegetation and to benefit from their respective advantages, i.e., larger coverage and higher resolution at the bank from oblique view and higher resolution to recognize distant GCPs from top view. Yet, a single oblique path could be utilized to optimize resources with improvements to the adopted approach (see previous Sections). In addition, other factors are relevant aiming at reducing errors as much as possible, such as choosing non-reflective materials for GCP targets, flying on overcast and bright days, using automated GCP identification algorithms, etc.

In our approach, GCPs were manually identified in the photographs based on three concentric geometries: the inner and outer circles of the CD reflective area and the tile perimeter. The centre of these geometries was estimated depending on the camera-GCP distance, which defined the target resolution, and the light reflexion intensities of each case. Then, fast flipping through photo focusing on single GCP at a time (with the PageUp / PageDown keys in PhotoScan) helped to adjust the estimation of the target centre, which turned consistent the location of the GCP among all camera views. The errors introduced during the target identification affect the georeferentiation, and although errors may compensate if not systematically biased, their influence is higher the narrower the cross-sectional GCP span due to the rotation tendency of elongated domains. This source of error may be reduced with wider GCP patterns than the used in our approach, but also through other improvements, such as automated identification of GCP and Monte Carlo tests (James et al., 2017), to identify more accurately the GCP and optimize the number of GCP and minimize DSM errors.

Finally, camera settings were automatic, being adjusted by the light conditions during the flight. But, the camera could be manually set to optimally capture the texture of the bank scarp. In this respect, GCP targets should ideally have a similar reflecting surface to project to the camera a similar amount of light for their later identification. A tilted target would also contribute to have similar reflecting conditions after comparable orientations to the bank with respect to the sun. Furthermore, if the date of the survey campaign is flexible, then overcast but bright days are advisable whenever possible. This is to avoid overexposed or underexposed bank areas due to direct sun light and shades that result in lower image textures within each of these zones and thus in a lower number of detected image features (James and Robson, 2012; Gómez-Gutierrez et al., 2014a). For example, note in the central photo of Fig. 13 that there were no textural differences due to shades but only due to the bank surface.

## 5.4 Surveying bank erosion

Sequential surveys allowed to capture different phases of the erosion cycle (Fig. 12a), which demonstrates that quantitative detection of processes is feasible. Previous studies on bank erosion proved the capabilities of SfM for post-event analysis (Prosdocimi et al., 2015), e.g. representing block deposition, or for 2.5D bank retreat quantification (Hamshaw et al., 2017),

whereas herein all erosion phases were sequentially captured, demonstrating the 3D potentialities over the complete process of erosion. Of course, the ability to monitor banks at the process scale depends on the time interval with which the method can re-survey the exposed part of banks and will only cover pre- and post-flood conditions. The survey frequency and the duration of a full cycle of erosion determine the temporal resolution with which the development of processes is captured. Then, the bank retreat rate of each case determines the necessary frequency of surveys to capture erosion processes within a

single cycle. Bank erosion rates naturally depend on each site, after different river sizes, hydraulic conditions, bank materials, etc. In the presented study site erosion rates varied enormously (Fig. 3), but still the performed eight surveys within a year successfully captured bank processes within a single erosion cycle in areas of fast retreat such as section 4.

The study site with a regulated water level and recently restored actively eroding banks was a perfect example for the application of this technique, because banks were exposed and erosion rates were compatible with the proposed average

sampling frequency of six weeks. For other types of rivers, where erosion mainly occurs during floods when banks are not exposed, this method would allow measuring pre- and post-event conditions only. Given the high resolution achieved, the method is applicable to all river sizes. However, due to the accuracy obtained, the application is only advised in cases where bank retreat is larger than approximately 30 cm between consecutive surveys.

Erosion processes happening at small spatial scales, such as weathering, would be hardly or not measurable with the

precision achieved in this investigation.. For this, other methods are already available, for instance TLS and boat-based laser scanning, that provide higher precisions (mm before registration errors, e.g., O'Neal and Pizzuto, 2011) and comparable resolutions (cm, e.g., Heritage and Hetherington, 2007). In addition, close-range terrestrial photogrammetry can also offer the necessary precision for such endeavours, e.g., from a tripod (Leyland et al., 2015) or a pole on the near-bank area (Bird et al., 2010), at the expense of covering shorter bank lengths. Another alternative are erosion pins, which may also provide

higher accuracies, yet with point resolution.

UAV-SfM appears a suitable survey method for both process identification and volume quantification in bank erosion studies, given the decimetre precision range with 3 cm RMSE and the 3D high resolution achieved with a low-cost UAV. As Resop and Hession (2010) suggested, high-resolution three-dimensional capabilities offer great possibilities when spatial variability of retreat is critical compared to traditional cross-profiling methods. In addition, the reduced deployment

time of UAVs in the field is advantageous in relation to cross-profiling, while it also improves identification of complex bank features (Fig. 11) and volume computations as other 3D high-resolution techniques (O'Neal and Pizzuto, 2011). Nonetheless, UAV-SfM require longer post-processing times at the office, which should not be underestimated (Westoby et al., 2012; Passalacqua et al., 2015).

This technique remains low-cost compared to TLS or MLS, for which it is more convenient for cases where roughness is beyond the scale of interest, and target bank lengths are smaller than 3000 m. This would approximately be the longest distance for a single UAV flight in our case study. For longer reaches, MLS would then compete with UAV-SfM from a practical perspective, since more than one survey/flight would be needed. However, all TLS, MLS and UAV-SfM would have limitations to survey the bank surface in presence of dense bank vegetation (Hamshaw et al., 2017). In these cases, ALS provides an alternative, albeit with significant lower resolution and higher costs (Slatton et al., 2007).

For large river extents, i.e., several kilometres, Grove et al. (2013) showed that process inference is possible combining ALS with high-resolution aerial photography, two techniques that are typically applied for eroded volume estimations and bank migration (Khan and Islam, 2003; Lane et al., 2010; De Rose and Basher, 2011; Spiekermann et al., 2017). In that work, the scale of the river (banks higher than 6 m) allowed a spatial resolution of 1 m to capture features that together with photo inspection provided information on mass failure type and fluvial entrainment. To date, UAV-SfM covers smaller extents (Passalacqua et al., 2015), but provides much higher resolutions, allowing for process identification (such as undermining) and more precise volume computations (see Fig. 11 for profile differences between ALS and SfM). For a similar (or higher) accuracy and resolution than those of UAV-SfM and large distances, boat-based laser scanning becomes an attractive, yet more expensive, solution.

## 6 Conclusion

This work evaluated the capability of Structure from Motion photogrammetry applied with low-cost UAV imagery to monitor bank erosion processes along a river reach. The technique's precision was investigated by comparison with GPS points and an airborne laser scanning. Vertical bank profiles were analysed to identify stages of erosion and infer processes. We used a low-cost UAV with a 12 MP built-in camera, flying 25 m from the least retreated bankline and 25 m above the floodplain level, which took oblique photographs with at least eight image overlaps of each bank point. The distribution of ground-control points across the floodplain avoided excessive linear errors from model rotation, keeping the accuracy within the target of 1/25 times the bank height. Thus, this GCP distribution and image network geometry generated through SfM a digital surface model with sufficient accuracy and resolution to recognize signatures of the different phases of the bank erosion cycle from bank profiles.

The accuracy of the DSM constructed with the SfM technique did not significantly increase with more than eight photo overlaps along a single oblique UAV track. The coverage of bank area behind bank toe vegetation, on the other hand, was increased by adding a vertically oriented perspective, albeit without a significant accuracy increase. As a result, banks were surveyed with 2 cm resolution and a 10 cm elevation precision, whose mean was 1 cm and standard deviation 3 cm (~1:1400 relative to camera-object distance, in line with previous SfM topographic applications). This accuracy was confirmed along the river reach with airborne laser scanning, although the latter overestimated elevations over bank slopes. Higher SfM errors were observed in areas beyond the extent of ground-control points, showing that control points should

also be placed outside the monitoring reach and close to the bankline. Furthermore, the GCP distribution across the floodplain proved very important to prevent model rotation along GCP axes, so a second line of GCP located further inland is recommended, together with proper targets, to reduce model errors as much as possible.

This investigation demonstrates the capabilities of a low-cost UAV to monitor banks at the process scale, while covering a middle-size river reach of 1.2 km long in a single campaign. The combination of UAV and Structure from Motion photogrammetry can provide relevant information of the spatial structure of bank erosion processes, and with sufficient frequency of acquisition, represent the temporal evolution of morphological processes within the erosion cycle. This method can also be used to compute eroded volumes throughout different phases of the cycle and analyse the contribution of each mechanism to overall retreats. The applied technique is most suitable when measuring bank lengths not exceeding 3000 m, and its flexibility, fast deployment and high resolution are especially convenient for surveying highly irregular banks. While this method can survey the full cycle of erosion, and not only pre- and post-event conditions, its main limitations are dense riparian vegetation and high water levels, as for most survey techniques.

**Data availability**

The data utilized in this work will be publicly available at the 4TU repository after the publication of this manuscript.

**Author contribution**

The field campaigns, data processing, data analysis and discussion of results were carried out by the first author. The writing of the manuscript was done by the first and second authors. The team authors provided expert opinion, refinement of paper writing and figures, and ideas that improved the final quality of this work.

**Acknowledgements**

This study was part of the NCR RiverCare research programme, funded by NWO/STW (project number 13516). We would like to thank Jaap van Duin and Ruben Kunz for their assistance during the field campaigns. We are grateful to Hans Bakker from Rijkswaterstaat for timely sharing the ALS data.

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

**Table 1**. Number of photographs and overlaps for the tests

|  | Test 1a | Test 1b | Test 1c | Test 1d | Test 2 | Test 3 | Test 4 | Test 5 |
|---|---|---|---|---|---|---|---|---|
| Track 1 | 293 | 147 | 73 | 37 |  | 147 |  | 293 |
| Track 2 |  |  |  |  | 232 | 232 |  | 232 |
| Track 3 |  |  |  |  |  |  | 232 | 232 |
| Track 4 |  |  |  |  |  |  | 232 | 232 |
| Min. overlaps | 16 | 8 | 4 | 2 | 7 | 15 | 26 | 49 |
| Max. overlaps | 20 | 10 | 5 | 2 | 7 | 17 | 26 | 53 |

**Table 2**. Mean and standard deviation of elevation differences between SfM DSMs and GPS points. Colour intensity indicates the deviation from zero value with minimum/maximum of ±0.13 m.

| Surface | Error (m) | Test 1a | Test 1b | Test 1c | Test 1d | Test 2 | Test 3 | Test 4 | Test 5 |
|---------|-----------|---------|---------|---------|---------|--------|--------|--------|--------|
| All grounds | Mean | -0.01 | 0.00 | 0.00 | 0.04 | 0.03 | -0.01 | -0.05 | 0.00 |
|  | Std. dev. | 0.03 | 0.03 | 0.04 | 0.07 | 0.03 | 0.03 | 0.05 | 0.03 |
| Grassland | Mean | 0.02 | 0.01 | 0.01 | 0.02 | 0.02 | 0.01 | 0.01 | 0.02 |
|  | Std. dev. | 0.03 | 0.03 | 0.04 | 0.04 | 0.03 | 0.02 | 0.02 | 0.02 |
| Bank | Mean | 0.00 | 0.01 | -0.01 | -0.01 | 0.05 | 0.01 | -0.03 | 0.01 |
|  | Std. dev. | 0.04 | 0.03 | 0.07 | 0.13 | 0.03 | 0.03 | 0.04 | 0.03 |
| Terrace | Mean | -0.02 | -0.02 | 0.00 | 0.06 | 0.03 | -0.02 | -0.09 | -0.01 |
|  | Std. dev. | 0.02 | 0.03 | 0.03 | 0.06 | 0.03 | 0.02 | 0.03 | 0.02 |

**Table 3**. Mean and standard deviation of elevation differences between SfM, ALS and RTK GPS. Colour intensity indicates the deviation from zero value with minimum/maximum of ±0.09 m.

| Subtraction | | All grounds | Grassland | Bank | Terrace |
|---|---|---|---|---|---|
| SfM - ALS (m) | Mean | -0.01 | 0.01 | -0.06 | -0.04 |
| | Std. dev. | 0.03 | 0.02 | 0.06 | 0.03 |
| ALS - GPS (m) | Mean | 0.01 | 0.01 | 0.01 | 0.01 |
| | Std. dev. | 0.05 | 0.03 | 0.09 | 0.05 |
| SfM - GPS (m) | Mean | -0.01 | 0.01 | 0.01 | -0.02 |
| | Std. dev. | 0.03 | 0.02 | 0.03 | 0.02 |