# Peer review of "A low-cost technique to measure bank erosion processes along middle-size river reaches"

_Earth Surface Dynamics, 2018_

## Referee Comment (RC1) · P. Carbonneau (Referee) · 5 Apr 2018

General Comments

This work attempts to deploy established SfM techniques over a 1.2 km rver bank characterised by bays and pseudo-headlands and with numerous vertical faces along the banks. The work does present some points of interest but the authors display an understanding of SfM-photogrammetry which is average/good and as a result seem to have missed many important points. The general pitch and justification of novelty for this paper is weak. This is expressed in the second sentence of the abstract stating 'Yet, no technique provides low-cost and high-resolution to survey small-scale bank processes along a river reach'. This is quite simply not true and is ultimately self-defeating as a

statement. The authors have deployed a widely used commercial drone and processed the data with an equally widely used commercial SfM package usng the standard workflow. There is no new technique in this work. The authors have cited other work that has delivered similar resolutions at slightly smaller scales or that have worked at lower resolutions at larger scales. Therefore the work occupies a very small niche of cm-scale resolution work over a 1.2km scale length. The workflow and techniques used by the authors are not at all new, they have merely benefited from better drone flight durations thus allowing them to cover a 1.2km river reach with repeated operations. In itself, this is not a sufficient justification for publication. However, the work does address an important and interesting problem. Using drone-based SfM for repeated coverage of a 1km river corridor, especially a very linear one as shown here, presents some very specific challenges. Moreover, when this reach presents some vertical surfaces prone to failure, additional photogrammetric challenges must be addressed. Unfortunately, the authors did not seem to realise that this was the key point and challenge of their work and, in addition to the misplaced pitch mentioned above, the data acquisition and processing approach is very sub-optimal for this specific problem.

Overall, I do think the data presented here has potential for publication without additional fieldwork, but some very significant revisions will be required which include both new analysis and re-writes of many sections. However, even if publishable without additional fieldwork, the workflow presented here is definitely not the optimal approach to survey long and highly linear river corridors and this will need to be clearly outlined in a new discussion.

Specific Comments

The key challenge in this work is the deployment of drone-based SfM over a very linear river reach characterised with near-vertical faces. This challenge can be understood if we consider the type of errors present in SfM point clouds. This is the main area where the authors understanding of SfM needs to improve. The error of georeferenced point clouds produced from SfM chan be partitionned in linear, non-linear and random

components. Linear errors affect the point cloud as a rigid block and can be expressed in terms of translation errors, rotation errors and scaling errors. Non-linear errors are often caused by camera calibration problems and are manifest by warps and curvature effects that distort the geometry of the point cloud. One notable error in the authors understanding is that the seem to think that camera calibration errors can cause overall rotation of the block as 1 rigid body. This is not the case. Camera calibration errors cause errors such as the now famous dish effect. However rigid block rotation errors are caused by errors in the 7-parameter Helmert transform used to scale, rotate and translate a relative point cloud to georeferenced coordinates. The parameters of the Helmert transform are calculated by least squares regression of the control data, if the control data is highly co-linear, the lest squares regression could converge on a false solution that is rotated around the axis formed by the line of control points. This is not the same as the non-linear camera calibration errors. It is also different from random errors are localised errors (often expressed as elevation errors) that are generally not spatially correlated and represent the classic concept of precision. In this work, the authors have been rigorous about possible effects of camera calibration and small scale noise, but they have completely missed possible rotation errors caused by the geometry of their case study.

From a photogrammetric perspective, there are 2 challenges posed by this case study. First, it is a highly linear reach with a very high length:width ration. Second, the presences of vertical faces will require highly oblique views. It is the first challenge that the authors have missed. There are 2 main problems in the data acquisition plan. First, the location of the GCPs is almost co-linear. As stated above,this means that numerical solutions to the georeferencing of the model (via the Helmert transform) will have a degree of equifinality around a family of solutions that rotate around this co-linear axis of GCP points. The addition of 2-3 points perhaps 50 meters inland would have reduced the co-linearity of the model. The authors need to consider the cross-stream footprint of their GCP points relative to the errors in the RTK GPS and in the human error associated to locating the GCP in an image (see below) and make a case that

the model is stable. In the future, the authors must seek to distribute their GCPs in a very non-colinear pattern that, as much as possible, occupies the full X, Y and Z extent of the area covered by imagery. Second, the choice of flight patterns that are lines parallel to the shore does not help this situation. A possible option would have been to fly the drone in a much wider pattern that goes further inland and off-shore. However, in this case, the relatively high error of the drone GPS would have required a fairly wide (in the cross-stream direction) flight pattern in order for the drone GPS data to make a beneficial contribution to the rotational stability of the model. Ultimately, the entire data acquisition setup proposed here is prone to delivering models that will have a tendency to present rotation errors where the entire point model is tilted with respect to an axis that is parallel to the shore line. This is a very significant weakness of this workflow. With this consideration in mind, it is very worrying that the authors have chosen to cut the data and have selected a portion of the point cloud that is near the GCP axis. In figure 7, the authors need to show the readers all the available data. Additionally, if they do choose to cut some peripheral data, some objective criteria must be chosen. And the moment, the choice of area seems subjective and does not give the reader confidence that the authors have not cherry-picked the part of their point cloud with the least error. I note that in figure 9, cross sections 1,2 and 4 do seem to have a rotational effect. The authors will need to demonstrate their current GCP setup does prevent rotation or return to the field with a better, wider GCP arrangement.

Technical Corrections

Abstract. Whilst Westoby et al 2012 did use hyphens when writing Structure-from-motion, this is an error . In the computer vision domain, where SfM was invented, hyphens are not used and so it is corectly written as Structure from Motion.

Section 2.2 This section does not cover the needed material to adress this case study. Need more on GCP distribution and on how a point cloud is georefenrced. See Fonstad et al 213 or Carbonneau and Dietrich 2017 (both aldready cited) for details

P7 line 7. By default, Cloud Compare computes differences along the Z axis. Did you use the M3C2 module which computes differences along the surface normals? Please clarify.

P8 line 13. Please add a photo of your target. If you have a 12cm diameter circle in the centre, then how do you accurately place the GCP in Photoscan? Are you using some machine assisted algorithm? At 25m, the P4 camera will give you 1.3 cm pixels at nadir. This means that you target centre could be almots 10 pixels wide. How do you find the exact centre and so benefit fully from the accuracy of the RTK GPS? Note that errors at this stage, combined with your co-linear steup of GCPS could contribute greatly to rotational errors of the whole point cloud.

P10 Figure 3. This is not a good view since scales are hard to determine. Better use a side view and a top view, both in orthometric perspectives so that a scale bar can be added.

Also, the choice of linear flight paths (here called tracks) parallel to the shore is again highly sub-optimal. This will only contribute to possible rotation errors. A grid pattern with multiple views would have been much more stable.

Figure 4. Please overlay the image footprints.

P11 From here you only use vertical error estimations to characterise method sucess. But as stated above, you could have other linear errors affecting the model. I note that the error distributions are bimodal with a dip for the number of errors in the 0 bin. This is consistent with a block rotation where few points (along the line of GCPs) are exactly corect. Many are either too high or too low. But this is not a vertical error in the photgrammetry process, it is the effect of rotation.

P13, Figure 7 Before you decide to crop data, you must show all the data. If you do crop, please select an objective criteria. e.g. 100 m buffer around each GCP. At the moment, the data looks manually cropped to variable distances away from the GCPs.

A more rigorous approach is needed.

Patrice Carbonneau

---

## Referee Comment (RC2) · Anonymous Referee #2 · 13 May 2018

The paper covers mapping a 1.2 km long river bank using a low-cost drone and structure from motion, and a comparison of this method with airborne laser scanning and RTK-GPS data. I do find this to be an interesting topic relevant for those working with morphology, environmental assessment and measurements in rivers, and it shows the potential of using drones and photogrammetry to gain data for river analysis. The comparison between different methods is also useful and relevant for other similar data collection efforts.

The paper is introduced in the abstract mostly as an example of developing a low-cost drone and the SfM method to measure the bank processes. The authors have employed a commercially available drone and commercial available software for this process, which in practice means that you already have a low-cost technique readily

available. This is then applied here to a new problem, and I think the manuscript should be focused on what is new here from the many other applications of the same combination of drone/software.

Issues related to the SfM procedure is covered in detail in another comment, but in line with the previous comment I would also like to ask for some more discussion in the paper on:

- The linear placement of the GCPs and the effect this might have on the point cloud. The placement of the control points and the effect this has on accuracy beyond the control points, and this could be combined to a more through discussion. - Issues related to the selection of flight paths only in parallel with the river bank that was measured.

I also think it would be good to show the targets used to define the GCP (what was the size of the tile?), and maybe also a picture showing how these looked in the images from the drone and how they were identified in Photoscan since this is relevant for the accuracy.

Further on GCPs, on P12 there is a discussion on the GCP identification between tracks. It would be particularly interesting to see some more info on how well you think the accuracy of the GCP identification from pictures were for Track 1 were you see the GCP tiles at an angle in all pictures. To what extent do the identification of the tile centre influence the results.

P7 – Was the grid from the ALS automatically generated from the scanner software or did you make it from the 10 points pr. square meter measured by the instrument? If it was automatic, could generating a finer grid improve the ALS results?

P8 – Did you use automatic or manual settings for shutter speed, ISO, etc.?

P10 - Figure 3: A top-view perspective could be added to improve the understanding of camera positions.

P10 – Line11: How was the removal of trees etc. done?

P11 – Table 2: Can you give a short explanation for the colour scale in the caption? Do the colour change for every 0.01 meters? I also think grouping mean and std.dev. together like it is done for the "all grounds" would be more readable also for Grassland, Bank and Terrace.

P13 – Figure 7 is hard to read. Is it possible to divide it into a panel with different sections of the bank in each panel to improve the readability of the figure?

P13 – Figure 8. I assume this is based on results after the removal of the data outside the GCP limit. How is the GCP limit defined?

P16 Figure 10a: Do I understand it correctly that the change of elevation of the flood plain is mainly caused by development of the grass?

P16: Figure 10. Would it be possible to illustrate the development here by also showing the images of at least some of the observations? The data presented in figure 10 is useful, but since there is both images and DTMs it would be interesting to see these processes and the basis for the development of figure 10.

P17-line 29: Can't you just write "A RMSE of 2.8 cm"?

P18 – Line19: Do you think GCP's at the bank toe would reduce the error? With reference to the previous comment on GCP placement and the further discussion on page 18, what was the rationale for the selected choice of GCP locations?

A last question just out of curiosity. Was the flight done in autonomous mode or under control? What regulations govern the use of small drones for research purposes in this area? I understand that this varies between countries and could be an issue in planning similar st

---

## Author Comment (AC1) · 25 Jun 2018

P. Carbonneau (Referee #1):

General Comments

RC: This work attempts to deploy established SfM techniques over a 1.2 km river bank characterised by bays and pseudo-headlands and with numerous vertical faces along the banks. The work does present some points of interest but the authors display an understanding of SfM-photogrammetry which is average/good and as a result seem to have missed many important points. The general pitch and justification of novelty for this paper is weak. This is expressed in the second sentence of the abstract stating 'Yet, no technique provides low-cost and high-resolution to survey small-scale bank

processes along a river reach'. This is quite simply not true and is ultimately self-defeating as a statement. The authors have deployed a widely used commercial drone and processed the data with an equally widely used commercial SfM package using the standard workflow. There is no new technique in this work. The authors have cited other work that has delivered similar resolutions at slightly smaller scales or that have worked at lower resolutions at larger scales. Therefore the work occupies a very small niche of cm scale resolution work over a 1.2km scale length. The workflow and techniques used by the authors are not at all new, they have merely benefited from better drone flight durations thus allowing them to cover a 1.2km river reach with repeated operations. In itself, this is not a sufficient justification for publication. However, the work does address an important and interesting problem. Using drone-based SfM for repeated coverage of a 1km river corridor, especially a very linear one as shown here, presents some very specific challenges. Moreover, when this reach presents some vertical surfaces prone to failure, additional photogrammetric challenges must be addressed. Unfortunately, the authors did not seem to realise that this was the key point and challenge of their work and, in addition to the misplaced pitch mentioned above, the data acquisition and processing approach is very sub-optimal for this specific problem. Overall, I do think the data presented here has potential for publication without additional fieldwork, but some very significant revisions will be required which include both new analysis and re-writes of many sections. However, even if publishable without additional fieldwork, the workflow presented here is definitely not the optimal approach to survey long and highly linear river corridors and this will need to be clearly outlined in a new discussion.

AC: We thank the referee for his critical and constructive comments on the UAV-SfM technique, which has helped improving the manuscript and the analysis of the specific challenges this case study proposes for the UAV-SfM application. The manuscript addresses two aspects which have been clarified and considered in the revised analysis. First, the application of an available technique to a new setting with particular characteristics and processes. The topography of the bank area has an unusual threedimensional complexity, it is tortuous and presents exposed undermined profiles, which overall lays on a quasi-linear domain, i.e., a straight river reach. We were aware of the latter but certainly did not address it as a novelty nor analysed the rotational tendency of the model. Second, the work provides evidence of a sufficiently accurate data acquisition approach to measure bank erosion processes, without the intention of achieving an optimal solution. Yet, we do realize that this should be thoroughly discussed in the manuscript in light of the referee's comments. In addition, the fact that now it is clear for us that the SfM camera calibration phase cannot re-adjust the linear transformation, provided the opportunity to improve the manuscript towards recommending a more robust UAV-SfM approach, as well as analysing and highlighting the particular challenges this setting offered for the UAV-SfM application. We also realize that the pitch was not accurate and should be phrased again. The spirit of the motivation lays on the application of a readily available technique to measure bank erosion at the process scale, which has unique characteristics compared to previously used methods for that aim, such as a combination of low-cost, fast deployment in the field, and high 3D resolution. Section 2.2 described past experiences with other techniques. Moreover, we compared the results with other two methods and discussed them in the broader context of other available techniques to put into perspective the convenience and disadvantages of UAV-SfM to measure bank erosion processes. We have, nevertheless, also focused the manuscript on the specific challenges this case study proposes for the UAV-SfM technique, addressing the rotational tendency of such linear domain, and assessing the performance of the adopted GCP distribution with parallel UAV paths.

Specific Comments

RC: The key challenge in this work is the deployment of drone-based SfM over a very linear river reach characterised with near-vertical faces. This challenge can be understood if we consider the type of errors present in SfM point clouds. This is the main area where the authors understanding of SfM needs to improve. The error of georeferenced point clouds produced from SfM chan be partitionned in linear, non-linear and

random components. Linear errors affect the point cloud as a rigid block and can be expressed in terms of translation errors, rotation errors and scaling errors. Non-linear errors are often caused by camera calibration problems and are manifest by warps and curvature effects that distort the geometry of the point cloud. One notable error in the authors understanding is that the seem to think that camera calibration errors can cause overall rotation of the block as 1 rigid body. This is not the case. Camera calibration errors cause errors such as the now famous dish effect. However rigid block rotation errors are caused by errors in the 7-parameter Helmert transform used to scale, rotate and translate a relative point cloud to georeferenced coordinates. The parameters of the Helmert transform are calculated by least squares regression of the control data, if the control data is highly co-linear, the lest squares regression could converge on a false solution that is rotated around the axis formed by the line of control points. This is not the same as the non-linear camera calibration errors. It is also different from random errors are localised errors (often expressed as elevation errors) that are generally not spatially correlated and represent the classic concept of precision. In this work, the authors have been rigorous about possible effects of camera calibration and small scale noise, but they have completely missed possible rotation errors caused by the geometry of their case study. From a photogrammetric perspective, there are 2 challenges posed by this case study. First, it is a highly linear reach with a very high length:width ration. Second, the presences of vertical faces will require highly oblique views. It is the first challenge that the authors have missed. There are 2 main problems in the data acquisition plan. First, the location of the GCPs is almost co-linear. As stated above,this means that numerical solutions to the georeferencing of the model (via the Helmert transform) will have a degree of equifinality around a family of solutions that rotate around this co-linear axis of GCP points. The addition of 2-3 points perhaps 50 meters inland would have reduced the co-linearity of the model. The authors need to consider the cross-stream footprint of their GCP points relative to the errors in the RTK GPS and in the human error associated to locating the GCP in an image (see below) and make a case that the model is stable. In the future, the authors

must seek to distribute their GCPs in a very non-colinear pattern that, as much as possible, occupies the full X, Y and Z extent of the area covered by imagery. Second, the choice of flight patterns that are lines parallel to the shore does not help this situation. A possible option would have been to fly the drone in a much wider pattern that goes further inland and off-shore. However, in this case, the relatively high error of the drone GPS would have required a fairly wide (in the cross-stream direction) flight pattern in order for the drone GPS data to make a beneficial contribution to the rotational stability of the model. Ultimately, the entire data acquisition setup proposed here is prone to delivering models that will have a tendency to present rotation errors where the entire point model is tilted with respect to an axis that is parallel to the shore line. This is a very significant weakness of this workflow. With this consideration in mind, it is very worrying that the authors have chosen to cut the data and have selected a portion of the point cloud that is near the GCP axis. In figure 7, the authors need to show the readers all the available data. Additionally, if they do choose to cut some peripheral data, some objective criteria must be chosen. And the moment, the choice of area seems subjective and does not give the reader confidence that the authors have not cherry-picked the part of their point cloud with the least error. I note that in figure 9, cross sections 1,2 and 4 do seem to have a rotational effect. The authors will need to demonstrate their current GCP setup does prevent rotation or return to the field with a better, wider GCP arrangement.

AC: As previously indicated, we assumed it was possible to re-adjust the linear transformation during the camera calibration step, and missed an analysis of possible rotation errors. We have added a new section 4.3 to analyze the linear rotation of the model, evaluating linear trends of elevation errors across the axis of potential rotation. We also included discussions on linear errors in section 5.1 and on the adopted workflow on a new section 5.3, including considerations on GCP distribution, target visualization and UAV paths. Regarding Figure 7, the cropping of the floodplain boundary was done to avoid the bush lines across the floodplain (visible in the background aerial photo) to prevent comparisons over vegetated areas. Also, the domain had been cropped at
the terrace end, before a narrow sloped strip because it was considered beyond the target area (the bank). Now, all available data has been plotted in the revised Figure 7 (whose extent was already visible in Figure 3). In addition, Figure 7 now has another panel showing the signed elevation differences between SfM and ALS for further analyses. The new Section 4.3 shows that there is a slight model rotation, resulting in an absolute elevation difference between the least and most retreated bank scarps of 4 cm. This has been discussed in Section 5.1 in the context of the other error sources. Also, the new Section 5.3 discusses the adopted workflow and recommends improvements for future works. The magnitude of the model rotation cannot be appreciated in the metre scale of Figures 9 and 10. These Figures show that the DSM was sufficiently accurate to measure bank erosion processes. This implies both sufficient accuracy and resolution at the bank area to quantify the three phases of the erosion cycle and the respective processes. We have added a quantitative reference to assess the performance of the method in this regard.

Technical corrections:

RC: Abstract. Whilst Westoby et al 2012 did use hyphens when writing Structure-from-motion, this is an error . In the computer vision domain, where SfM was invented, hyphens are not used and so it is corectly written as Structure from Motion.

AC: The hyphens have been removed in all sections.

RC: Section 2.2 This section does not cover the needed material to adress this case study. Need more on GCP distribution and on how a point cloud is georefenrenced. See Fonstad et al 213 or Carbonneau and Dietrich 2017 (both aldready cited) for details

AC: An explanation of the particular challenges of this case study for SfM has been added in Section 2.2. Also, a description of error sources and model georeferentiation has been added to Section 3.3.

RC: P7 line 7. By default, Cloud Compare computes differences along the Z axis. Did

you use the M3C2 module which computes differences along the surface normals? Please clarify.

AC: The two computations P7 line 7 refers to were done with the cloud/cloud distance tool of CloudCompare. The distances between the 129 GPS points and the ALS grid were computed with a local 2.5D Delaunay triangulation of the latter. Then, the vertical component was obtained whose statistical results were presented in Table 3. The, ALS – SfM comparison was done between the nearest points of the respective clouds. This has been clarified in Section 3 of the manuscript.

RC: P8 line 13. Please add a photo of your target. If you have a 12cm diameter circle in the centre, then how do you accurately place the GCP in Photoscan? Are you using some machine assisted algorithm? At 25m, the P4 camera will give you 1.3 cm pixels at nadir. This means that you target centre could be almots 10 pixels wide. How do you find the exact centre and so benefit fully from the accuracy of the RTK GPS? Note that errors at this stage, combined with your co-linear steup of GCPS could contribute greatly to rotational errors of the whole point cloud.

AC: The centre of the targets was manually identified, without any machine assistance. The identification of the target centre relied on three concentric geometries, which were respectively used depending on the camera-GCP distance and the specific light conditions of each case. The smallest target was the CD inner circle with approximately 3.5 cm of diameter. This was used whenever visible. For the furthest targets and in those cases where the whole CD or tile were reflecting too much light to the camera, the CD or tile centres were estimated based on the shape of their boundaries. Later, fast flipping through photo focusing on single GCP at a time (with the PageUp / PageDown keys in PhotoScan) helped to adjust the estimation of the target centre. This procedure turned consistent the location of the GCP among all camera views. Yet, this does not prevent introducing errors when identifying GCP target locations. A photo of the target has been added to Figure 3, together with two more panels showing how a target is seen from the UAV paths 1 and 2. A new Section 5.3.3 clarifies the GCP identification

procedure and the effects it has on accuracy.

RC: P10 Figure 3. This is not a good view since scales are hard to determine. Better use a side view and a top view, both in orthometric perspectives so that a scale bar can be added.

AC: Two scales were added in the perspective view of the DSM as references (for the corridor width and bank scarp height). Figure 3 has been expanded with two more panels showing an orthometric top view from Photoscan and an schematic side view with the UAV positions (since the sideview in Photoscan turned confusing with too many photographs).

RC: Also, the choice of linear flight paths (here called tracks) parallel to the shore is again highly sub-optimal. This will only contribute to possible rotation errors. A grid pattern with multiple views would have been much more stable.

AC: We have included a discussion on UAV paths in Section 5.3.2.

RC: Figure 4. Please overlay the image footprints.

AC: It has been done.

RC: P11 From here you only use vertical error estimations to characterize method success. But as stated above, you could have other linear errors affecting the model. I note that the error distributions are bimodal with a dip for the number of errors in the 0 bin. This is consistent with a block rotation where few points (along the line of GCPs) are exactly corect. Many are either too high or too low. But this is not a vertical error in the photgrammetry process, it is the effect of rotation.

AC: Thanks for this interesting observation. We have plotted the elevation errors at the rotational plane to analyse the signs and distances to the rotation axis. Indeed, this tendency is confirmed showing that errors at the floodplain and the terrace not only have respective positive and negative biases (as already presented in Table 2, column for Test 3), but also present a consistent trend across the rotational axis. This is then most

likely caused by a small model rotation. We also acknowledge the role of vegetation cover (grass over the floodplain) as a source of overestimating ground elevations, and we do not discard non-linear effects on the terrace (beyond GCP limits), as possible errors present in the DSM that result in the achieved overall model accuracy. Overall, we agree with the analysis on the basis of the results, and do not discard the influence of other error sources that may contribute to enhance the linear trend. The analysis of linear rotation of the model is in a new section 4.3, and the results are discussed in section 5.1.

RC: P13, Figure 7 Before you decide to crop data, you must show all the data. If you do crop, please select an objective criteria. e.g. 100 m buffer around each GCP. At the moment, the data looks manually cropped to variable distances away from the GCPs. A more rigorous approach is needed.

AC: The crop criterion along the floodplain had the intention to avoid the lines of brushes that lay across it (visible from the aerial photograph in the background). This shape was the result of cropping the LIDAR data prior to the comparison with the SfM point cloud. For a more robust approach, the ALS data has been cut along the boundaries of the SfM dense point cloud (visible in Figure 3), which are bounded by the area photographed from the nadiral UAV view (track 2). This criterion results in a overlapping domain between ALS and SfM data with an approximate constant width of 42m. Figure 7 now shows all available data.

---

## Author Comment (AC2) · 25 Jun 2018

Anonymous Referee #2:

RC: The paper covers mapping a 1.2 km long river bank using a low-cost drone and structure from motion, and a comparison of this method with airborne laser scanning and RTK-GPS data. I do find this to be an interesting topic relevant for those working with morphology, environmental assessment and measurements in rivers, and it shows the potential of using drones and photogrammetry to gain data for river analysis. The comparison between different methods is also useful and relevant for other similar data collection efforts. The paper is introduced in the abstract mostly as an example of developing a lowcost drone and the SfM method to measure the bank processes.

[Figure]

The authors have employed a commercially available drone and commercial available software for this process, which in practice means that you already have a low-cost technique readily available. This is then applied here to a new problem, and I think the manuscript should be focused on what is new here from the many other applications of the same combination of drone/software. Issues related to the SfM procedure is covered in detail in another comment, but in line with the previous comment I would also like to ask for some more discussion in the paper on: - The linear placement of the GCPs and the effect this might have on the point cloud. The placement of the control points and the effect this has on accuracy beyond the control points, and this could be combined to a more through discussion. - Issues related to the selection of flight paths only in parallel with the river bank that was measured.

AC: We thank the referee for his comments which helped improving the manuscript. A new section 5.3 has been added to discuss the novel challenges that the case study presents for the UAV-SfM application. A discussion on the linear placement of GCPs and its effect on the point cloud has been done in section 5.3.2. The placement of GCPs and their effect on the accuracy beyond GCPs has been discussed in sections 5.1 and 5.3.1. The main effect is the non-linear distortion beyond GCPs that produces the "dome" shape of the model, visible in Figure 7 at the extremes of the reach. The selection of parallel UAV paths and their role in the model registration has been discussed in the new section 5.3.2.

RC: I also think it would be good to show the targets used to define the GCP (what was the size of the tile?), and maybe also a picture showing how these looked in the images from the drone and how they were identified in Photoscan since this is relevant for the accuracy.

AC: A photo of a target has been added in Figure 3, together with two more panels that show how it is seen from UAV tracks 1 and 2. The plaques were 40x40 cm (this has been added in section 3.2). The targets were manually identified in Photoscan, and with the help of PageUp/PageDown keys we achieved consistency between all photos

focusing on every target. This was useful since the target resolution varied with the sensor-GCP distance and some plaques at oblique perspectives reflected too much light to identify the CD (usually from track 1 and far away from the cross-section where the target was). In these cases, the background texture with the plaque boundaries were also used as references to locate the plaque's centre. If the plaque's centre was not clear enough to be identified, the GCP at that photo was discarded. The explanation and discussion of this criterion has been added to the new section 5.3.3.

RC: Further on GCPs, on P12 there is a discussion on the GCP identification between tracks. It would be particularly interesting to see some more info on how well you think the accuracy of the GCP identification from pictures were for Track 1 were you see the GCP tiles at an angle in all pictures. To what extent do the identification of the tile centre influence the results.

AC: Indeed, from the inclined angle of track 1 a smaller target area was captured, proportionally diminished by the cosine of the viewing angle with respect to the normal of the plane in which the GCP laid (say alfa). Following the previous explanation, this created more light reflection especially from the largest distances (oblique in two directions), but logically there also was a lower resolution to identify the target centres. The error introduced by a coarser resolution translated into elevation errors, and these directly affect rotational errors. On the other hand, the lateral view helped to compensate for this, since this elevation errors decreased with the cosine of alfa. Yet, positioning errors in transverse direction are relevant to quantify bank erosion rates, so these should be kept as low as possible, for instance, with inclined targets. A clarification on the influence of GCP identification on accuracy and a recommendation have been added in the new section 5.3.1.

RC: P7 – Was the grid from the ALS automatically generated from the scanner software or did you make it from the 10 points pr. square meter measured by the instrument? If it was automatic, could generating a finer grid improve the ALS results?

AC: We did not process the raw data because we did not have access to it. We only worked with the 0.5x0.5m grid provided to us. The grid resolution is in principle limited by the footprint size of the laser beam on the ground and the number of points per square metre. The footprint size was approximately 0.16m (one third of the grid resolution), from the beam divergence angle and the flight height. So, a finer grid could have been produced, for instance a 0.33x0.33 m grid from the 10 points/m$^2$ to approximately have one cell per each point. In this sense, the ALS results could have probably improved to capture more irregularities on the ground, which for banks proved essential. Still, even with a finer resolution, ALS can only survey 2.5D without capturing undermined profiles. These clarifications were added to sections 3 and 5.2.

RC: P8 – Did you use automatic or manual settings for shutter speed, ISO, etc.?

AC: We used automatic camera settings. Some considerations on this respect have been added in section 5.3.3.

RC: P10 – Figure 3: A top-view perspective could be added to improve the understanding of camera positions.

AC: A top view has been added to Figure 3, with two lines representing UAV paths 1 and 2-4.

RC: P10 – Line11: How was the removal of trees etc. done?

AC: The removal of vegetation from the point cloud was manually done in PhotoScan, simply by rotating the DSM and selecting those points above the surrounding ground level. Trees were easy to remove and some bushes required finding a correct perspective not to erase points from the ground.

RC: P11 – Table 2: Can you give a short explanation for the colour scale in the caption? Do the colour change for every 0.01 meters? I also think grouping mean and std.dev. together like it is done for the "all grounds" would be more readable also for Grassland, Bank and Terrace.

AC: For the mean values, the colour scale highlights the deviation from the zero value. For the naked eye it is possible to distinguish colour changes every 0.005 m, for the given range of values between -0.13 and +0.13 m (Table 2). In the case of the standard deviation, the colour scale highlights the deviation from the zero value too, but for a range of values between 0.00 and 0.13 m. Table 2 has been rearranged according to the suggestion and a description of the colour scale has been added in the caption.

RC: P13 – Figure 7 is hard to read. Is it possible to divide it into a panel with different sections of the bank in each panel to improve the readability of the figure?

AC: Figure 7 has been divided into two panels to increase the data visibility.

RC: P13 – Figure 8. I assume this is based on results after the removal of the data outside the GCP limit. How is the GCP limit defined?

AC: That is correct. The limit was defined by cross-sectional straight lines at the centre of the extreme GCPs. These lines have been added to Figure 7.

RC: P16 Figure 10a: Do I understand it correctly that the change of elevation of the flood plain is mainly caused by development of the grass?

AC: That is correct. The floodplain grass is mowed every year in October and usually grows during spring and summer time. The development of grass can be now observed in a new Figure 13 showing UAV photos from track 1.

RC: P16: Figure 10. Would it be possible to illustrate the development here by also showing the images of at least some of the observations? The data presented in figure 10 is useful, but since there is both images and DTMs it would be interesting to see these processes and the basis for the development of figure 10.

AC: A new Figure 13 has been added to show the development through three consecutive surveys at km. 153.9 (cross section 4), with a short description of processes linked to Figure 10 (called Fig. 12 in the new manuscript).

RC: P17-line 29: Can0t you just write "A RMSE of 2.8 cm"?

AC: Of course, thanks.

RC: P18 – Line19: Do you think GCP0s at the bank toe would reduce the error? With reference to the previous comment on GCP placement and the further discussion on page 18, what was the rationale for the selected choice of GCP locations?

AC: We think that non-linear errors could be reduced if GCPs were placed at the bank toe, but at the same time, there was no clear evidence of this type of error at the bank face. Regarding linear errors, the contribution to avoid or reduce rotation errors around the GCP axis would be minor, especially compared to wider horizontal GCP distributions across the floodplain. This is because the bank was roughly 3.5 metres high and its horizontal extent was also relatively short, compared to the floodplain that could allow for much larger cross-sectional distances to stabilize the DSM. What is more, GCPs were not systematically placed at the bank toe throughout the surveys for practical reasons. First, it was faster just to place GCPs on the floodplain than to descend to the bank toe, measure the coordinates, climb up, fly the UAV and sometimes recover the plaques afterwards (not all of them were left on the field to save them from vandalism). Second, we tried placing plaques at the bank toe but ship waves were able to take them away. Therefore, the GCPs were located only over the floodplain, following the bankline. With sufficient cross-sectional GCP footprint and photo overlaps, the model should be stable enough to measure bank erosion, as in the case study presented. The criterion has been indicated in section 3.2, and a discussed in section 5.3.2.

RC: A last question just out of curiosity. Was the flight done in autonomous mode or under control? What regulations govern the use of small drones for research purposes in this area? I understand that this varies between countries and could be an issue in planning similar st

AC: The flight was done in autonomous mode with the software UgCS. There are regulations regarding no-fly zones that are typically close to airports (see

https://kadata.kadaster.nl/dronekaart/) and pilot licenses are now necessary to fly UAVs. It is true that it can be an issue to fly UAVs so it is necessary to check beforehand what the local regulations are and sometimes ask for specific permissions before a mission.

---

## Referee Report (RR1)

**A low-cost technique to measure bank erosion processes along middle-size river reaches**

Gonzalo Duró[1], Alessandra Crosato[1,2], Maarten G. Kleinhans[3], Wim S. J. Uijttewaal[1]

[1] Department of Hydraulic Engineering, Delft University of Technology, PO Box 5048, 2600 GA Delft, the Netherlands
5  [2] Department of Water Engineering, IHE-Delft, PO Box 3015, 2601 DA Delft, the Netherlands
[3] Department of Physical Geography, Utrecht University, PO Box 80115, 3508 TC Utrecht, the Netherlands

*Correspondence to*: Gonzalo Duró (G.Duro@tudelft.nl)

We investigate the capabilities of Structure from Motion (SfM) photogrammetry applied with imagery from an Unmanned Aerial Vehicle (UAV) to measure bank erosion processes in middle-size rivers. This technique offers a unique set of characteristics compared to previously used methods to monitor banks, such as high resolution, low-cost and relatively fast deployment in the field. We analyse a 1.2 km restored bank of the Meuse River with complex vertical scarps laying on a straight reach, features that present specific challenges to the UAV-SfM application. We surveyed eight times within a year, combining different photograph perspectives and overlaps to identify an effective UAV flight. The accuracy of the Digital Surface Models (DSMs) was evaluated with RTK GPS points and an Airborne Laser Scanning (ALS) of the whole reach. An oblique perspective with eight photo overlaps and 20 m of cross-sectional ground-control point (GCP) distribution were sufficient to achieve the relative precision to observation distance of ~1:1400 and 3 cm RMSE, complying with the required accuracy. A complementary nadiral view increased coverage behind bank toe vegetation. The GCP footprint across the floodplain proved critical to avoid rotation of straight elongated domains, so improvements to the adopted approach are recommended. Sequential DSMs captured signatures of the erosion cycle such as mass failures, slump-block deposition, and bank undermining. Although this technique requires low water levels and banks without dense vegetation as many others, it is an inexpensive and fast-in-the-field alternative to survey reach-scale riverbanks in sufficient resolution to quantify bank retreat and identify morphological features of the bank failure and erosion processes.

Keywords:    Riverbank erosion monitoring, erosion cycle, restoration, Unmanned Aerial Vehicles (UAV), Structure from Motion (SfM).

**1 Introduction**

Bank erosion is a fundamental process in morphologically active river systems, and much research has been devoted to understanding, quantifying and modelling it from disciplines such as engineering, geomorphology, geology and ecology. River bank erosion involves interconnected physical, chemical and biological processes (e.g., Hooke, 1979; ASCE, 1998; Rinaldi and Darby, 2008), resulting in a complex phenomenon that is difficult to thoroughly understand and predict (e.g.,

**Summary of Comments on Blank**

**Page: 1**

Number: 1     Author: patca     Subject: Sticky Note     Date: 23/08/2018 14:58:49
This title is not very indicative of the main points of the paper.  I would also dispute the 'low-cost' claim since you use an RTK-GPS which is not low-cost.
Why not have some mention of sub-vertical scarps and high length to width areas?

Number: 2     Author: patca     Subject: Sticky Note     Date: 23/08/2018 14:51:11
mid-sized

Number: 3     Author: patca     Subject: Cross-Out  Date: 23/08/2018 14:51:51

Number: 4     Author: patca     Subject: Sticky Note     Date: 23/08/2018 14:53:49
This sentence is out of place.  Clearly written just to ward off review comments, but it does not make sense in the abstract for the other readers.

Number: 5     Author: patca     Subject: Sticky Note     Date: 23/08/2018 14:55:45
Given that your results are based on RTK-GPS ground control, it is not inexpensive.

[revised manuscript text omitted]

**Number: 1**     Author: patca     Subject: Sticky Note     Date: 23/08/2018 15:02:43

best use the term 'quality' precision has a specific meaning.

**Number: 2**     Author: patca     Subject: Sticky Note     Date: 23/08/2018 15:02:04

precision and accuracy are separate concepts. precision is variance around the truth and accuracy is systematic bias away from truth.

**Number: 3**     Author: patca     Subject: Sticky Note     Date: 23/08/2018 15:03:44

do you mean neither? As in all these methods do NOT have good resolution? Or do you mean that both DO have good resolution?

**Number: 4**     Author: patca     Subject: Sticky Note     Date: 23/08/2018 15:04:45

Could add some papers by JP Bailly and Paul Kinzel

**Number: 5**     Author: patca     Subject: Inserted Text     Date: 21/08/2018 12:00:31

interpretation

**Number: 6**     Author: patca     Subject: Sticky Note     Date: 21/08/2018 12:01:34

slightly awkward to include ALS in a paragraph on phtoography. ALS was discussed above so this reference to ALS needs to be re-located.

**Number: 7**     Author: patca     Subject: Sticky Note     Date: 23/08/2018 15:07:27

if you mean 2D bank retreat, then this sentence needs to move up a bit because you have started talking about 3D measurements.

**Number: 8**     Author: patca     Subject: Sticky Note     Date: 21/08/2018 12:03:18

this is basically the same starting sentence as in the paragraph starting around line 12. This needs tightening up.

[revised manuscript text omitted]

**Page: 6**

Number: 1      Author: patca     Subject: Sticky Note     Date: 21/08/2018 12:06:38

paragraph needs regroupiong.  Probably best to follow chronological order of publications.

Number: 2      Author: patca     Subject: Sticky Note     Date: 23/08/2018 15:17:26

no. You have mis-read the work of Eltner.  There are certain guiding principles and it's mis-leading to imply otherwise.
This is also not a good citation for this snetrence.  Review papers often skip over high levels of technical detail to get a larger paper.  It is lazy scholarship to read 1 review paper and then skip reading the detailed papers listed in the review paper.  In this specific case, a comprehensive read of the works of Mike James is required.

Number: 3      Author: patca     Subject: Sticky Note     Date: 21/08/2018 12:09:35

You did not use the 'flexibility' you used the UAV.  Tighten up the language here.

the georeferentiation accuracy regarding the model rotation around GCP axis. Fourth, we searched for bank features in SfM-based profiles and analogous ones from ALS, and for signatures of erosion processes along sequential SfM surveys.

For the first step, the analysis of the minimum number of photographs needed to achieve the highest DSM precision, we compared the DSMs with RTK GPS measurements to quantify vertical accuracy. We took 129 points across eight profiles on 18-01-2017 (see Fig. 4) with a Leica GS14 RTK GPS, whose root mean square precisions according to the manufacturer specifications are 8 mm + 0.5 ppm in horizontal and 15 mm + 0.5 ppm in vertical directions. On the same date, we flew the UAV along the bank four times with different camera angles and perspectives. Eight photograph combinations were considered to derive 8 DSMs. Then, the comparisons were done with the elevation differences between the GPS points and the corresponding closest ones of the DSM point clouds (e.g., Westoby et al, 2012; Micheletti et al., 2015). We used CloudCompare software (Girardeau-Montaut, 2017) for these computations.

In the second step, we compared the selected DSM from the previous analysis with reach-scale survey technique, ALS, to analyse topographic differences over the whole river reach. The ALS was carried out on 17-01-2017 from an airplane at 300 meters above the ground level. The laser scanner, a *Riegl LMS-Q680i,* measured a minimum of 10 points per square metre with an effective pulse rate of 266 kHz. We did not have access to the raw data and used the automatically generated 0.5 m grid. We tested the ALS elevation precision against the 129 RTK GPS points using the vertical component of the closest distance to a local Delaunay triangulation of the ALS grid, due to the different resolutions between both datasets. Then, we computed the distances between the ALS grid points and the corresponding nearest ones of the DSM point cloud. We did both computations with the standard cloud/cloud distance tool of CloudCompare, distinguishing between surfaces of grassland, bare ground and bank.

Third, we analysed the DSM spatial stability with respect to the potential axis of rotation around the GCPs, which the case study laid over a narrow, elongated and straight domain. The GCPs distributed over the floodplain along the near-bank area defined the linear transformation from an arbitrarily scaled coordinate system to the real-world coordinates. In order to verify that the DSM was stable and the tendency to rotate around co-linear solutions did not affect the accuracy beyond the survey target, we computed a regression line with the GCPs to identify the potential axis of rotation for the DSM domain. Then, we projected onto the perpendicular rotational plane the DSM elevation errors corresponding to the GPS points and computed a second regression line to evaluate if there was a linear tendency that indicated a model rotation.

Fourth, we made profiles across six sections of dissimilar erosion rates to contrast the bank representations of i) the SfM DSM, ii) the triangulated ALS grid, and iii) the RTK GPS points. The profiles were computed with MATLAB using i) the Geometry Processing Toolbox (Jacobson et al., 2017) adapted to slice triangle meshes, ii) a linear interpolation across the triangulated ALS grid, and iii) a projection of the RTK GPS points onto the exact cross-section locations. Then, we identified and analysed a cross section over which sequential SfM-UAV surveys showed different stages of the erosion cycle, since the bank erosion cycle was used as a reference to distinguish between techniques capable of measuring at either the process or the cross-sectional scale.

**Number: 1**      Author: patca      Subject: Cross-Out   Date: 21/08/2018 12:10:23

**Number: 2**      Author: patca      Subject: Sticky Note      Date: 23/08/2018 15:23:13

acquired

**Number: 3**      Author: patca      Subject: Cross-Out   Date: 23/08/2018 15:23:01

**Number: 4**      Author: patca      Subject: Inserted Text      Date: 23/08/2018 15:23:23

data

**Number: 5**      Author: patca      Subject: Sticky Note      Date: 23/08/2018 15:24:23

second half of the sentence is not quite clear.

**Number: 6**      Author: patca      Subject: Sticky Note      Date: 23/08/2018 15:27:25

This does not make sense, there are an infinity of rotational planes around the GCP-axis

**Number: 7**      Author: patca      Subject: Sticky Note      Date: 23/08/2018 15:28:12

So you mean that you expect a linear increase of error as you move away from the axis?

**Number: 8**      Author: patca      Subject: Sticky Note      Date: 23/08/2018 15:31:25

I have to wonder why you are using 2D profiles after going through so much effort to produce a fully 3D DSM.  All the work you describe here could be done in CloudCompare with the point clouds.  You could use the M3C2 routine by Lague in order to check erosion along the surface normals.

**Number: 9**      Author: patca      Subject: Inserted Text      Date: 23/08/2018 15:28:48

extracted

[revised manuscript text omitted]

Number: 1    Author: patca    Subject: Sticky Note    Date: 21/08/2018 12:14:21
no that's not correct.  The user may not input these and their retrieval is more automated, but they remain critical.

Number: 2    Author: patca    Subject: Inserted Text    Date: 23/08/2018 15:39:47
existing.  Features need not be pre-defined (e.g. corners) to be matched.

Number: 3    Author: patca    Subject: Cross-Out  Date: 23/08/2018 15:40:37

Number: 4    Author: patca    Subject: Sticky Note    Date: 23/08/2018 15:41:26
Not exclusively.  See works on direct georeferencing by Turner; Carbonneau and others.  EvenJames et al 2017 discussed DG.

Number: 5    Author: patca    Subject: Inserted Text    Date: 23/08/2018 15:41:40
georeferencing.

Number: 6    Author: patca    Subject: Sticky Note    Date: 21/08/2018 12:15:16
not just in SfM, traditional photorgamm4etry does this too.

We used Agisoft PhotoScan software to process the imagery. For a successful photo alignment from different UAV tracks (Table 1), the camera yaw, pitch and roll recorded during the UAV flight were necessary inputs. For this step we used three GCPs along the reach, two at the extremes and one at the middle, all close to the bank and easily visible from tracks 1 and 2. These approximate orientations and a priori known ground points helped obtaining a consistent sparse point cloud of the bank along the entire reach. The resulting camera positions and orientations of the photo alignment are visible in Fig. 3a, evidencing the UAV tracks. This figure also shows the DSM textured with colours from the photographs, in which the green area on the left side with white patches corresponds to the floodplain partially covered with snow (see also Fig. 3d-f) and the right brownish area is the terrace at the bank toe, with snow remains as well.

[Figure]

**Figure 3:** a) Camera positions and orientations in perspective view. The digital surface model shows the low-water condition during January 2017, which exposed a terrace at the bank toe. b) Cross-sectional scheme of UAV paths. c) Ceramic plaque with CD as ground control point on the floodplain. d) GCP in photograph from track 1. e) Same GCP from track 2. f) Top view of DSM with UAV track 1 in blue and tracks 2-4 in yellow.

After obtaining the sparse point cloud, we marked the remaining 15 GCPs (Fig. 4). Then, we refined the camera parameters by minimizing the sum of GCP reprojection and misalignment errors. This camera optimization adjusts the estimated point cloud by reducing non-linear deformations. Once the dense point cloud was computed, we removed the points outside the area of interest, as well as those points at the water surface, tree canopies and individual bushes at the floodplain. Finally, the point cloud was triangulated and interpolated to generate a triangle mesh. This mesh consisted of a non-monotonic surface that was later processed in MATLAB to plot 2D cross sections.

Number: 1    Author: patca    Subject: Sticky Note    Date: 21/08/2018 12:15:57

no.  PS can determine yaw, pitch and roll.

Number: 2    Author: patca    Subject: Sticky Note    Date: 23/08/2018 15:48:00

What do you mean marked? as in measured?

Why not use ALL the points in alignement?  Can you cite some manual or literature to support your fragmented usage approach?  I have never seen this elsewhere.

Number: 3    Author: patca    Subject: Sticky Note    Date: 23/08/2018 15:44:41

You mean that you used the optimization button in PS?

Number: 4    Author: patca    Subject: Inserted Text    Date: 23/08/2018 15:45:12

TIN

Number: 5    Author: patca    Subject: Sticky Note    Date: 23/08/2018 15:46:27

Need to specify meshing parameters?  Did you use 'Arbitrary'?  In fot, then your entire premise of bank overhangs cannot work, but if you did, I  would like a discussion of the processing implications becasue arbitrary meshes consume a very large amount of RAM.

[revised manuscript text omitted]

Number: 1    Author: patca    Subject: Sticky Note    Date: 23/08/2018 16:09:01

This new figure now shows that you have a focal length problem. Your high areas are too high, and the low areas are too low. The vertical scale of the domain is not correct and this is due to the focal. Calibration of the focal is improved by a diversity of elevation in the survey data.

Number: 2    Author: patca    Subject: Sticky Note    Date: 22/08/2018 15:31:40

You missed a key point here: based on figure 7b, you have a doming deformation in the DOWNSTREAM direction. This effect is added to a mis-calibration of the focal length that results in a slightly incorrect vertical scale.

[revised manuscript text omitted]

Number: 1      Author: patca      Subject: Sticky Note      Date: 22/08/2018 15:53:30

Severe over-usage of the word Yet, you need to cut about 80% of these.

Number: 2      Author: patca      Subject: Sticky Note      Date: 23/08/2018 16:00:05

This should be a much more complete statement giving a summary of flight requirements in terms of number of images per km (or some linear unit) and flying height/distance.

Number: 3      Author: patca      Subject: Cross-Out    Date: 23/08/2018 16:00:57

Number: 4      Author: patca      Subject: Inserted Text      Date: 23/08/2018 16:01:56

co-linear

Number: 5      Author: patca      Subject: Inserted Text      Date: 23/08/2018 16:02:21

rotated

[revised manuscript text omitted]

---

## Author Response (AR2)

Dear Richard Gloaguen,

We have modified the manuscript following your suggestions. We did our best to make clear that the goal is to measure banklines and not to propose a new method. We also took into consideration most of the comments by Rev#1 to improve the readability and quality of the manuscript.

Kind regards,
Gonzalo Duró

[revised manuscript text omitted]